# DIVERGENCE-AWARE FEDERATED SELF-SUPERVISED LEARNING

**Weiming Zhuang**[1,3]**, Yonggang Wen**[2]**, Shuai Zhang**[3]
[1]S-Lab, NTU, Singapore [2]NTU, Singapore [3]SenseTime Research
`weiming001@e.ntu.edu.sg,ygwen@ntu.edu.sg,zhangshuai@sensetime.com`

## ABSTRACT

Self-supervised learning (SSL) is capable of learning remarkable representations from centrally available data. Recent works further implement federated learning with SSL to learn from rapidly growing decentralized unlabeled images (e.g., from cameras and phones), often resulted from privacy constraints. Extensive attention has been paid to SSL approaches based on Siamese networks. However, such an effort has not yet revealed deep insights into various fundamental building blocks for the federated self-supervised learning (FedSSL) architecture. We aim to fill in this gap via in-depth empirical study and propose a new method to tackle the non-independently and identically distributed (non-IID) data problem of decentralized data. Firstly, we introduce a generalized FedSSL framework that embraces existing SSL methods based on Siamese networks and presents flexibility catering to future methods. In this framework, a server coordinates multiple clients to conduct SSL training and periodically updates local models of clients with the aggregated global model. Using the framework, our study uncovers unique insights of FedSSL: 1) stop-gradient operation, previously reported to be essential, is not always necessary in FedSSL; 2) retaining local knowledge of clients in FedSSL is particularly beneficial for non-IID data. Inspired by the insights, we then propose a new approach for model update, Federated Divergence-aware Exponential Moving Average update (FedEMA). FedEMA updates local models of clients adaptively using EMA of the global model, where the decay rate is dynamically measured by model divergence. Extensive experiments demonstrate that FedEMA outperforms existing methods by 3-4% on linear evaluation. We hope that this work will provide useful insights for future research.

## 1 INTRODUCTION

Self-supervised learning (SSL) has attracted extensive research interest for learning representations without relying on expensive data labels. In computer vision, the common practice is to design proxy tasks to facilitate visual representation learning from unlabeled images (Doersch et al., 2015; Noroozi & Favaro, 2016; Zhang et al., 2016; Gidaris et al., 2018). Among them, the state-of-the-art SSL methods employ contrastive learning that uses Siamese networks to minimize the similarity of two augmented views of images (Wu et al., 2018; Chen et al., 2020a; He et al., 2020; Grill et al., 2020; Chen & He, 2021). All these methods heavily rely on the assumption that images are centrally available in cloud servers, such as public data on the Internet.

However, the rapidly growing amount of decentralized images may not be centralized due to increasingly stringent privacy protection regulations (Custers et al., 2019). The increasing number of edge devices, such as street cameras and phones, are generating a large number of unlabeled images, but these images may not be centralized as they could contain sensitive personal information like human faces. Besides, learning representations from these images could be more beneficial for downstream tasks deployed in the same scenarios (Yan et al., 2020). A straightforward method is to adopt SSL methods for each edge, but it results in poor performance (Zhuang et al., 2021a) as decentralized data are mostly non-independently and identically distributed (non-IID) (Li et al., 2020a).

Federated learning (FL) has emerged as a popular privacy-preserving method to train models from decentralized data (McMahan et al., 2017), where clients send training updates to the server instead

of raw data. The majority of FL methods, however, are not applicable for unsupervised representation learning because they require fully labeled data (Caldas et al., 2018), or partially labeled data in either the server or clients (Jin et al., 2020a; Jeong et al., 2021). Recent studies implement FL with SSL methods that are based on Siamese networks, but they only focus on a single SSL method. For example, FedCA (Zhang et al., 2020a) is based on SimCLR (Chen et al., 2020a) and FedU (Zhuang et al., 2021a) is based on BYOL (Grill et al., 2020). These efforts have not yet revealed deep insights into the fundamental building blocks of Siamese networks for federated self-supervised learning.

In this paper, we investigate the effects of fundamental components of federated self-supervised learning (FedSSL) via in-depth empirical study. To facilitate fair comparison, we first introduce a generalized FedSSL framework to embrace existing SSL methods that differ in building blocks of Siamese networks. The framework comprises of a server and multiple clients: clients conduct SSL training using Siamese networks — an online network and a target network; the server aggregates the trained online networks to obtain a new global network and uses this global network to update the online networks of clients in the next round of training. FedSSL primarily focuses on the cross-silo FL where clients are stateful with high availability (Kairouz et al., 2019).

We conduct empirical studies based on the FedSSL framework and discover important insights of FedSSL. Among four popular SSL methods (SimCLR (Chen et al., 2020a), MoCo (He et al., 2020), BYOL (Grill et al., 2020), and SimSiam (Chen & He, 2021), FedBYOL achieves the best performance, whereas FedSimSiam yields the worst performance. More detailed analysis uncover the following unique insights: 1) Stop-gradient operation, essential for SimSiam and BYOL, is not always essential in FedSSL; 2) Target networks of clients are essential to gain knowledge from online networks; 3) Keeping local knowledge of clients is beneficial for performance on non-IID data.

Inspired by the insights, we propose a new approach, Federated Divergence-aware Exponential Moving Average update (FedEMA) [1] , to address the non-IID data problem. Specifically, instead of updating online networks of clients simply by the global network, FedEMA updates them via exponential moving average (EMA) of the global network, where the decay rate of EMA is measured by the divergence of global and online networks dynamically. Extensive experiments demonstrate that FedEMA outperforms existing methods in a wide range of settings. We believe that important insights from this study will shed light on future research. Our main contributions are threefold:

- We introduce a new generalized FedSSL framework that embraces existing SSL methods based on Siamese networks and presents flexibility catering to future methods.

- We conduct in-depth empirical studies of FedSSL based on the framework and discover deep insights of the fundamental building blocks of Siamese networks for FedSSL.

- Inspired by the insights, we further propose a new model update approach, FedEMA, that adaptively updates online networks of clients with EMA of the global network. Extensive experiments show that FedEMA outperforms existing methods in a wide range of settings.

## 2 RELATED WORK

**Self-supervised Learning** In computer vision, self-supervised learning (SSL) aims to learn visual representations without any labels. Discriminative SSL methods facilitate learning with proxy tasks (Pathak et al., 2016; Noroozi & Favaro, 2016; Zhang et al., 2016; Gidaris et al., 2018). Among them, contrastive learning (Oord et al., 2018; Bachman et al., 2019) has become a promising principle. It uses Siamese networks to minimize the similarity of two augmented views (positive pairs) and maximize the similarity of two different images (negative pairs). These methods are either contrastive or non-contrastive ones: *contrastive* SSL methods require negative pairs (Chen et al., 2020a; He et al., 2020) to prevent training collapse; *non-contrastive* SSL methods (Grill et al., 2020; Chen & He, 2021) are generally more efficient as they maintain remarkable performances using only positive pairs. However, these methods do not perform well on decentralized non-IID data (Zhuang et al., 2021a). We analyze their similarities and variances and propose a generalized FedSSL framework.

**Federated Learning** Federated learning (FL) is a distributed training technique for learning from decentralized parties without transmitting raw data to a central server (McMahan et al., 2017).

---

[1]Intuitively, FedSSL is analogous to a SuperClass in object-oriented programming (OOP), then FedEMA is a SubClass that inherits FedSSL and overrides the model update method.

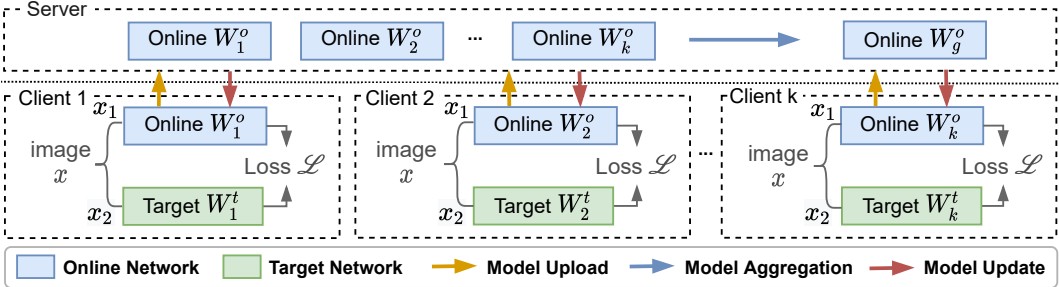

Figure 1: Overview of federated self-supervised learning (FedSSL) framework. It comprises an end-to-end training pipeline with four steps: 1) Each client $k$ conducts local training on unlabeled data $\mathcal{D}_k$ with Siamese networks — an online network $W_k^o$ and a target network $W_k^t$; 2) After training, client $k$ uploads $W_k^o$ to the server; 3) The server aggregates them to obtain a new global network $W_g^o$; 4) The server updates $W_k^o$ of client $k$ with $W_g^o$.

Among many studies that address the non-IID data challenge (Zhao et al., 2018; Li et al., 2020b; Wang et al., 2020; Zhuang et al., 2021c), Personalized FL (PFL) aims to learn personalized models for clients (Tan et al., 2021). Although some PFL methods interpolate global and local models (Hanzely et al., 2020; Mansour et al., 2020; yuyang deng et al., 2021), our proposed FedEMA differ in the motivation, application scenario, and measurement of the decay rate. Besides, the majority of existing works only consider supervised learning where clients have fully labeled data. Although recent works propose federated semi-supervised learning (Jin et al., 2020b; Zhang et al., 2020b; Jeong et al., 2021) or federated domain adaptation (Peng et al., 2020; Zhuang et al., 2021b), they still need labels in either the server or clients. This paper focuses on purely unlabeled decentralized data.

**Federated Unsupervised Learning**   Learning representations from unlabeled decentralized data while preserving data privacy is still a nascent field. Federated unsupervised representation learning is first proposed by van Berlo et al. (2020) based on autoencoder, but it neglects the non-IID data challenge. Zhang et al. (2020a) address the non-IID issue with potential privacy risk for sharing features. Although Zhuang et al. (2020) address the issue based on BYOL as our FedEMA, they do not shed light on why BYOL works best. Since SSL methods are evolving rapidly and new methods are emerging, we introduce a generalized FedSSL framework and deeply investigate the fundamental components to build up practical guidelines for the generic FedSSL framework.

## 3   AN EMPIRICAL STUDY OF FEDERATED SELF-SUPERVISED LEARNING

This section first defines the problem and introduces the generalized FedSSL framework. Using the framework, we then conduct empirical studies to reveal deep insights of FedSSL.

### 3.1   PROBLEM DEFINITION

FedSSL aims to learn a generalized representation $W$ from multiple decentralized parties for downstream tasks in the same scenarios. Each party $k$ contains unlabeled data $\mathcal{D}_k = \{\mathcal{X}_k\}$ that cannot be transferred to the server or other parties due to privacy constraints. Data is normally non-IID among decentralized parties (Li et al., 2020a); each party could contain only limited data categories (e.g., two out of ten CIFAR-10 classes) (Luo et al., 2019). As a result, each party alone is unable to obtain a good representation (Zhuang et al., 2021a). The global objective function to learn from multiple parties is $\min_w f(w) := \sum_{k=1}^{K} \frac{n_k}{n} f_k(w)$, where $K$ is the number of clients, and $n = \sum_{k=1}^{K} n_k$ is the total data amount. For client $k$, $f_k(w) := \mathbb{E}_{x_k \sim \mathcal{P}_k}[\tilde{f}_k(w; x_k)]$ is the expected loss over data distribution $\mathcal{P}_k$, where $x_k$ is the unlabeled data and $\tilde{f}_k(w; x_k)$ is the loss function.

### 3.2   GENERALIZED FRAMEWORK

We introduce a generalized FedSSL framework that empowers existing SSL methods based on Siamese networks to learn from decentralized data under privacy constraints. Figure 1 depicts the

end-to-end training pipeline of the framework. It comprises of three key operations: 1) Local Training in clients; 2) Model Aggregation in the server; 3) Model Communication (upload and update) between the server and clients. We implement and analyze four popular SSL methods — SimCLR (Chen et al., 2020a), MoCo (V1 (He et al., 2020) and V2 (Chen et al., 2020b)), SimSiam (Chen & He, 2021), and BYOL (Grill et al., 2020). Variances in Siamese networks of these methods lead to differences in executions in these three operations [2].

**Local Training** Firstly, each client $k$ conducts self-supervised training on unlabeled data $\mathcal{D}_k$ based on the same global model $W_g^o$ downloaded from the server. Regardless of SSL methods, clients train with Siamese networks — an online network $W_k^o$ and a target network $W_k^t$ for $E$ local epochs using cooresponding loss functions $\mathcal{L}$. We classify these SSL methods with two major differences (Figure 8 in Appendix A): 1) Only SimSiam and BYOL contain a *predictor* in the online network, so we denote their online network $W_k^o = (W_k, W_k^p)$, where $W_k$ is the online encoder and $W_k^p$ is the predictor; As for SimCLR and MoCo, $W_k^o = W_k$. 2) SimCLR and SimSiam share identical weights between the online encoder and the target encoder, so $W_k^t = W_k$. In contrast, MoCo and BYOL update the target encoder with EMA of the online encoder in every mini-batch: $W_k^t = mW_k + (1 - m)W_k^t$, where $m$ is the momentum value normally set to 0.99.

**Model Communication** After local training, client $k$ uploads the trained online network $W_k^o$ to the server and updates it with the global model $W_g^o$ after aggregation. Considering the differences of SSL methods, we upload and update encoders and predictors separately: 1) we upload and update the predictor when it presents in local training; 2) we follow the communication protocol Zhuang et al. (2021a) to upload and update only the online encoder $W_k$ when encoders are different.

**Model Aggregation** When the server receives online networks from clients, it aggregates them to obtain a new global model $W_g^o = \sum_{k=0}^{K} \frac{n_k}{n} W_k^o$. $W_g^o = (W_g, W_g^p)$ if predictor presents, otherwise $W_g^o = W_g$, where $W_g$ is the global encoder. Then, the server sent $W_g^o$ to clients to update their online networks. The training iterates these three operations until it meets the stopping conditions. At the end of the training, we use the parameters of $W_g^o$ as the generic representation $W$ for evaluation.

## 3.3 EXPERIMENTAL SETUP

We provide basic experimental setups in this section and describe more details in Appendix B.

**Datasets** We conduct experiments using CIFAR-10 and CIFAR-100 datasets (Krizhevsky et al., 2009). To simulate federated settings, we equally split a dataset into $K$ clients. We simulate non-IID data with label heterogeneity, where each client contains limited classes — $l = \{2, 4, 6, 8, 10\}$ number of classes for CIFAR-10 and $l = \{20, 40, 60, 80, 100\}$ for CIFAR-100. The setting is IID when each client contains 10 (100) classes for CIFAR-10 (CIFAR-100).

**Implementation Details** We implement FedSSL in Python using popular deep learning framework PyTorch (Paszke et al., 2017). To simulate federated learning, we train each client on one NVIDIA V100 GPU. These clients communicate with the server through NCCL backend. We use ResNet-18 (He et al., 2016) as default network for the encoders and present results of ResNet-50 in Appendix C. The predictor is a two-layer multi-layer perceptron (MLP). By default, we train for $R = 100$ rounds with $K = 5$ clients, $E = 5$ local epochs, batch size $B = 128$, learning rate $\eta = 0.032$ with cosine decay, and non-IID data $l = 2$ ($l = 20$) for CIFAR-10 (CIFAR-100).

**Linear Evaluation** We evaluate the quality of representations following linear evaluation (Kolesnikov et al., 2019; Grill et al., 2020) protocol. We first learn representations from the FedSSL framework. Then, we train a new linear classifier on the frozen representations.

## 3.4 ALGORITHM COMPARISONS

We benchmark and compare the SSL methods using the FedSSL framework. To denote the implementation of an SSL method, We add a prefix *Fed* to the name of the SSL method. For example, FedBYOL denotes using BYOL in the FedSSL framework.

---

[2]Intuitively, FedSSL is analogous to a SuperClass in OOP, then implementation of each method is a SubClass that inherits FedSSL and overrides methods of local training, model communication, and aggregation.

Table 1: Top-1 accuracy comparison of SSL methods using the FedSSL framework on non-IID CIFAR datasets. FedBYOL performs the best, whereas FedSimSiam performs the worst.

| Type | Method | CIFAR-10 (%) | CIFAR-100 (%) |
|---|---|---|---|
| Contrastive | FedSimCLR | $78.09 \pm 0.14$ | $55.58 \pm 0.13$ |
| | FedMoCoV1 | $78.21 \pm 0.04$ | $56.98 \pm 0.29$ |
| | FedMoCoV2 | $79.14 \pm 0.13$ | $57.47 \pm 0.65$ |
| Non-contrastive | FedSimSiam | $76.27 \pm 0.18$ | $48.94 \pm 0.22$ |
| | FedBYOL | $\mathbf{79.44 \pm 0.99}$ | $\mathbf{57.51 \pm 0.09}$ |

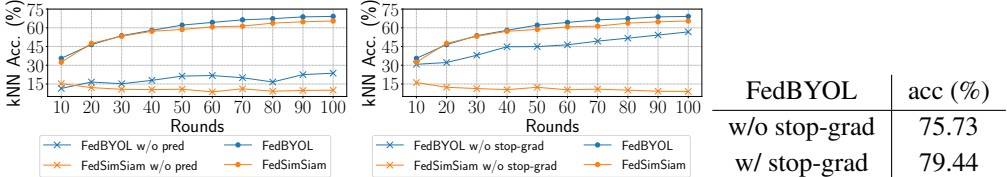

| FedBYOL | acc (%) |
|---|---|
| w/o stop-grad | 75.73 |
| w/ stop-grad | 79.44 |

Figure 2: Comparison of non-contrastive FedSSL methods with and without (w/o) predictor (pred) or stop-gradient (stop-grad) on non-IID CIFAR-10 dataset. Without predictor, both FedSimSiam and FedBYOL drops performance on kNN testing accuracy (left plot). Without stop-gradient, FedBYOL retains competitive results on kNN testing accuracy (middle plot) and linear evaluation (right table).

Table 1 compares linear evaluation results of these methods under the non-IID setting of CIFAR datasets. On the one hand, *contrastive* FedSSL methods obtain similar performances. As SimCLR is previously reported to need a large batch size (e.g., $B = 4096$) (Chen et al., 2020a), it is surprising that FedSimCLR obtains competitive results using the same batch size $B = 128$ as the others. On the other hand, the results of *non-contrastive* FedSSL methods have large variances: FedBYOL achieves the best performance, whereas FedSimSiam yields the worst performance. Since SimSiam is capable to learn as powerful representations as BYOL (Chen & He, 2021), as well as considering that non-contrastive methods are conceptually simpler and more efficient (Tian et al., 2021), we focus on non-contrastive methods and further investigate the effects of their fundamental components.

## 3.5 IMPACT OF FACTORS OF NON-CONTRASTIVE METHODS

This section analyzes the impact of fundamental components of non-contrastive FedSSL methods. From empirical studies, we obtain the following insights: 1) predictor is essential; 2) EMA and stop-gradient improves performances; 3) Local encoders should retain local knowledge of the non-IID data; 4) Target encoder should gain knowledge from the online encoder. Details are as followed.

**Predictor** is essential. Figure 2 (left plot) presents the kNN testing accuracy as a monitoring process for FedBYOL and FedSimSiam with and without predictors. Without predictors, both methods can barely learn due to collapse in local training. It affirms the vital role of predictor (Chen & He, 2021; Tian et al., 2021) even when learning from decentralized data.

**Stop-gradient** operation is previously indicated as an essential component for SimSiam and BYOL (Tian et al., 2021), but it is *not essential* for FedBYOL. Stop-gradient prevents stochastic gradient optimization on the target network. Figure 2 shows that FedSimSiam without stop-gradient collapses, whereas FedBYOL without stop-gradient still achieves competitive performance. It is because online and target encoders are significantly different in FedBYOL as the online encoder is updated by the global encoder every communication round. In contrast, SimSiam or FedSiamSiam share weights between online and target encoders, so removing stop-gradient leads to collapse.

**Exponential Moving Average (EMA)** is not essential, but it helps improve performance. Table 2 (first and second rows) shows that FedBYOL outperforms FedSimSiam at different levels of non-IID data, which is represented by {2, 4, 6, 8, 10} classes per client of CIFAR-10. EMA is the main difference between SimSiam and BYOL, indicating that EMA is helpful to improve performance. Based on these results, we further analyze the underlying impact of EMA on the encoders below.

Table 2: Top-1 accuracy comparison on various non-IID levels — the number of classes per client on the CIFAR-10 dataset. Update-both means updating both $W_k$ and $W_k^t$ with $W_g$.

| Method | # of classes per client (%) | | | | |
|---|---|---|---|---|---|
| | 2 | 4 | 6 | 8 | 10 (iid) |
| FedBYOL | 79.44 | 82.82 | 83.02 | 84.57 | 84.20 |
| FedSimSiam | 76.27 | 79.34 | 80.17 | 80.92 | 80.50 |
| FedBYOL, update-both | 74.50 | 78.77 | 83.02 | 84.56 | 83.80 |

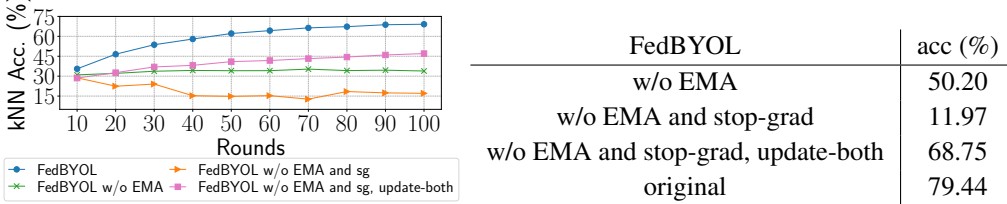

| FedBYOL | acc (%) |
|---|---|
| w/o EMA | 50.20 |
| w/o EMA and stop-grad | 11.97 |
| w/o EMA and stop-grad, update-both | 68.75 |
| original | 79.44 |

Figure 3: Comparison of FedBYOL without exponential moving average (EMA) and stop-gradient (sg) on the non-IID CIFAR-10 dataset. FedBYOL w/o EMA and sg can hardly learn, but updating both $W_k$ and $W_k^t$ with $W_g$ (update-both) enables it to achieve comparable results.

**Encoders** that retain local knowledge of non-IID data helps improve performance. EMA in Fed-BYOL allows the parameters of the online encoder to be different from the target encoder. As a result, the global encoder only updates the online encoder, not the target encoder. We hypothesize that retaining such local knowledge of data in the target encoder is beneficial especially when the data distribution is highly skewed. For comparison, we remove such local knowledge by updating both online and target encoders with the global model. Table 2 shows that FedBYOL with both encoders updated leads to lower performance than FedBYOL; It achieves results close to FedSimSiam when the data distribution is more skewed (2 or 4 classes per client). These results demonstrate the importance of keeping local knowledge in the encoders. Besides, the results of {6, 8, 10} classes per client also indicate the benefit of EMA.

**Target encoder** is essential to gain knowledge from the online encoder. Figure 3 shows that Fed-BYOL without EMA can merely learn, and FedBYOL without EMA and stop-gradient (sg) degrades in performance. In both cases, the target encoder is either never updated (w/o EMA) or is updated only through backpropagation (w/o EMA and sg) — not updated by the online encoder. On the other hand, we also identify that FedSSL methods in Table 1, which achieve competitive results, all update target encoders with knowledge of online encoders (the global encoder is the aggregation of the online encoder). We argue that target encoder is crucial to gain knowledge from the online encoder to provide contrastive targets. We further validate it by using the global encoder to update both online and target encoders when removing both EMA and stop-gradient. Figure 3 shows that such method improves performance and achieves comparable results.

## 4 DIVERGENCE-AWARE DYNAMIC MOVING AVERAGE UPDATE

Built on the FedSSL framework, we propose Federated Divergence-aware EMA update (FedEMA) to further mitigate non-IID data challenges. Since FedBYOL contains all components that help improve performance, we adopt it as the baseline and optimize the *model update* operation.

Non-IID data causes the global model to diverge from centralized training (Zhuang et al., 2021a). Inspired by the insight that retaining local knowledge of non-IID data helps improve performance, we propose to update the online network via EMA of the global network. Compared with FedBYOL that replaces the online network with the global network, FedEMA fuses local and global knowledge effectively through EMA update, where the decay rate of EMA is dynamically measured by model divergences. Figure 4 depicts our proposed FedEMA method. The formulation is as followed:

$$W_k^r = \mu W_k^{r-1} + (1-\mu)W_g^r, \tag{1}$$

$$W_k^{p,r} = \mu W_k^{p,r-1} + (1-\mu)W_g^{p,r}, \tag{2}$$

$$\mu = \min(\lambda \left\| W_g^r - W_k^{r-1} \right\|, 1), \tag{3}$$

where $W_k^r$ and $W_k^{p,r}$ are the online encoder and predictor of client $k$ at training round $r$; $W_g^r$ and $W_g^{p,r}$ are the global encoder and predictor; $\mu$ is the decay rate, measured by the divergence between global and online encoders; $\lambda$ is a scaler to adjust the level of model divergence, which is measured by calculating the $l_2$-norm of the global and online encoders. We summarize FedEMA in Algorithm 1. FedEMA can be regarded as a generalization of FedBYOL — they are the same when $\lambda = 0$.

Scaler $\lambda$ plays a vital role to adapt FedEMA for different levels of divergence caused by the data. The divergence between global and online encoders varies when the settings of federated learning change. For example, different degrees of non-IID settings would result in different divergences. Since characteristics of data are unknown before training as they are unlabeled, we propose a practical *autoscaler* to calculate a personalized $\lambda_k$ for each client $k$ automatically. The formula is $\lambda_k = \frac{\tau}{||W_g^{r+1} - W_k^r||}$, where $\tau \in [0,1]$ is the expected value of $\mu$ at round $r$. We calculate $\lambda_k$ only once at the earliest round $r$ that client $k$ is sampled for training. When the same set of clients are sampled for training, $\lambda_k = \frac{\tau}{||W_g^1 - W_k^0||}$ is calculated at round $r = 1$.

The intuition of FedEMA is to retain more local knowledge when divergence is large and incorporate more global knowledge when divergence is small. When model divergence is large, keeping more local knowledge is more beneficial for the non-IID data. Since the global network is the aggregation of online networks, representing global knowledge from clients. When divergence is small, adapting more global knowledge help improve model generalization. Since model divergence is larger at the start of training (Figure 6), it is practical to choose larger $\tau \in [0.5, 1)$; $\tau = 1$ is not considered because only local knowledge is used when $\tau = 1$. We use $\tau = 0.7$ by default in experiments.

---

**Algorithm 1** Our proposed FedEMA

1: **ServerExecution:**
2: Init $W_g^0$ and $W_g^{p,0}$, init $\lambda_k$ to be *null*
3: **for** *each round* $r = 0, 1, ..., R$ **do**
4:     $S_t \leftarrow$ (Selection of K clients)
5:     **for** *client* $k \in S_t$ *in parallel* **do**
6:         $W_k^r, W_k^{p,r} \leftarrow$ Client($W_g^r, W_g^{p,r}, r, \lambda_k$)
7:     $W_g^{r+1} \leftarrow \sum_{k \in S_t} \frac{n_k}{n} W_k^r$
8:     $W_g^{p,r+1} \leftarrow \sum_{k \in S_t} \frac{n_k}{n} W_k^{p,r}$
9:     **for** *client* $k \in S_t$ **do**
10:         $\lambda_k \leftarrow \frac{\tau}{||W_g^{r+1} - W_k^r||}$ **if** $\lambda_k$ is *null*
11: **Return** $W_g^R$
12: **Client** ($W_g^r, W_g^{p,r}, r, \lambda_k$):
13: **if** $\lambda_k$ is *null or* not selected in $r - 1$ **then**
14:     $W_k, W_k^t, W_k^p \leftarrow W_g^r, W_g^r, W_g^{p,r}$
15: **else**
16:     $\mu \leftarrow \min(\lambda_k \left\| W_g^r - W_k^{r-1} \right\|, 1)$
17:     $W_k \leftarrow \mu W_k^{r-1} + (1-\mu)W_g^r$
18:     $W_k^p \leftarrow \mu W_k^{p,r-1} + (1-\mu)W_g^{p,r}$
19: **for** *local epoch* $e = 0, 1, ..., E - 1$ **do**
20:     **for** $b \in \mathcal{B}$ *data batches with size* $B$ **do**
21:         $W_k^o \leftarrow W_k^o - \eta \nabla \mathcal{L}_{W_k^o, W_k^t}(W_k^o; b)$
22:         $W_k^t \leftarrow m W_k^t + (1-m)W_k$
23: **Return** $W_k^r, W_k^{p,r}$

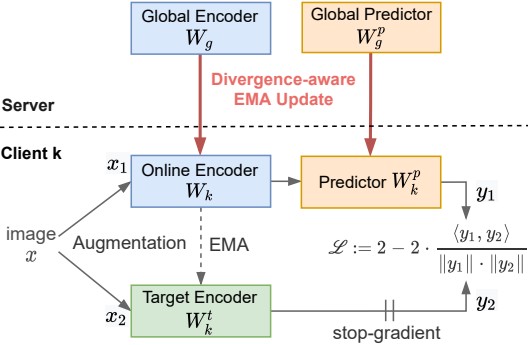

Figure 4: Illustration of our proposed Federated Divergence-aware Exponential Moving Average update (FedEMA). Compared with FedBYOL that simply updates the online network $W_k^o$ of client $k$ with the global network $W_g^o$, we propose to update them via EMA of the global network following Eqn 1 and 2, where the decay rate $\mu$ is dynamically measured the divergences between the online encoder $W_k$ and the global encoder $W_g$ (Eqn 3). The online network, $W_k^o = (W_k, W_k^p)$, is the concatenation of the online encoder $W_k$ and the predictor $W_k^p$.

Table 3: Top-1 accuracy comparison under linear probing on CIFAR datasets. Our proposed FedEMA outperforms all other methods. Full results are in Table 5.

| Method | CIFAR-10 (%) | | CIFAR-100 (%) | |
|---|---|---|---|---|
| | IID | Non-IID | IID | Non-IID |
| Standalone training | $82.42 \pm 0.32$ | $74.95 \pm 0.66$ | $53.88 \pm 2.24$ | $52.37 \pm 0.93$ |
| FedBYOL | $84.29 \pm 0.18$ | $79.44 \pm 0.99$ | $54.24 \pm 0.24$ | $57.51 \pm 0.09$ |
| FedU (Zhuang et al., 2021a) | $83.96 \pm 0.18$ | $80.52 \pm 0.21$ | $54.82 \pm 0.67$ | $57.21 \pm 0.31$ |
| FedEMA ($\lambda = 0.8$) | $\mathbf{85.59 \pm 0.25}$ | $\mathbf{82.77 \pm 0.08}$ | $\mathbf{57.86 \pm 0.15}$ | $\mathbf{61.21 \pm 0.54}$ |
| FedEMA (autoscaler, $\tau = 0.7$) | $\mathbf{86.26 \pm 0.26}$ | $\mathbf{83.34 \pm 0.39}$ | $\mathbf{58.55 \pm 0.34}$ | $\mathbf{61.78 \pm 0.14}$ |
| BYOL (Centralized) | $90.46 \pm 0.34$ | - | $65.54 \pm 0.47$ | - |

Table 4: Top-1 accuracy comparison on 1% and 10% of labeled data for semi-supervised learning on non-IID CIFAR datasets. FedEMA outperforms other methods. Full results are in Table 7.

| Method | CIFAR-10 (%) | | CIFAR-100 (%) | |
|---|---|---|---|---|
| | 1% | 10% | 1% | 10% |
| Standalone training | $61.37 \pm 0.13$ | $69.06 \pm 0.24$ | $21.37 \pm 0.73$ | $39.99 \pm 0.87$ |
| FedBYOL | $70.48 \pm 0.30$ | $76.95 \pm 0.46$ | $30.21 \pm 0.40$ | $47.07 \pm 0.14$ |
| FedU (Zhuang et al., 2021a) | $69.52 \pm 0.73$ | $77.06 \pm 0.55$ | $29.00 \pm 0.27$ | $46.67 \pm 0.06$ |
| FedEMA ($\lambda = 1$) | $\mathbf{72.78 \pm 0.66}$ | $\mathbf{79.01 \pm 0.30}$ | $\mathbf{32.49 \pm 0.22}$ | $\mathbf{49.82 \pm 0.36}$ |
| FedEMA (autoscaler, $\tau = 0.7$) | $\mathbf{73.44 \pm 0.22}$ | $\mathbf{79.49 \pm 0.34}$ | $\mathbf{33.04 \pm 0.23}$ | $\mathbf{50.48 \pm 0.11}$ |
| BYOL (Centralized) | $87.67 \pm 0.15$ | $87.89 \pm 0.05$ | $40.96 \pm 0.58$ | $56.60 \pm 0.33$ |

## 5 EVALUATION

This section follows the experimental setup in Section 3.3 to evaluate FedEMA in the linear evaluation and semi-supervised learning. We also provide ablation studies of important hyperparameters.

### 5.1 ALGORITHM COMPARISONS

To demonstrate the effectiveness of FedEMA, we compare it with the following methods: 1) standalone training, where a client learns independently using BYOL; 2) FedCA, which is proposed in Zhang et al. (2020a); 3) FedBYOL as the baseline; 4) FedU, which is proposed in (Zhuang et al., 2021a). Besides, we also present results of possible upper bounds that learn representations with centralized data using BYOL.

**Linear Evaluation** Table 3 shows that FedEMA outperforms other methods on different settings of CIFAR datasets. Specifically, the performance is more 3% higher than existing methods in most settings. Besides, our proposed autoscaler achieves similar results as $\lambda = 0.8$. More experiments on larger number of clients $K$ and random selection of clients are provided in Table 6 in Appendix C.

**Semi-supervised Learning** We also assess the quality of representations following the semi-supervised learning protocol (Zhai et al., 2019; Chen et al., 2020a) — we add a new two-layer MLP on top of the encoder and fine-tune the whole model with limited (1% and 10%) labeled data for 100 epochs. Table 4 indicates that FedEMA consistently outperforms other methods on non-IID settings of CIFAR datasets and our autoscaler outperforms manual-selected $\lambda = 1$.

### 5.2 ABLATION STUDIES

**Ablation on FedEMA** We analyze whether we need to update both online encoder (Eqn 1) and predictor (Eqn 2) in FedEMA. Figure 5 shows that updating only the encoder or predictor leads to better performance; only updating predictor also leads to faster convergence. Their combination re-

| Method | acc (%) |
|---|---|
| FedBYOL | 79.44 |
| FedEMA predictor only | 81.13 |
| FedEMA encoder only | 82.39 |
| FedEMA (ours) | 82.77 |

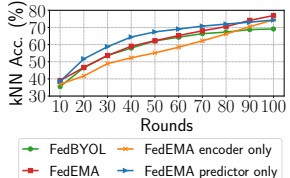

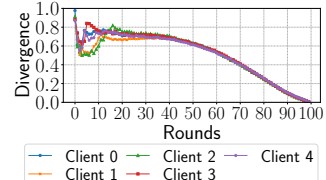

Figure 5: Ablation studies of FedEMA: applying EMA on either predictor or encoder leads to better performance on CIFAR-10.

Figure 6: Changes of divergence throughout training.

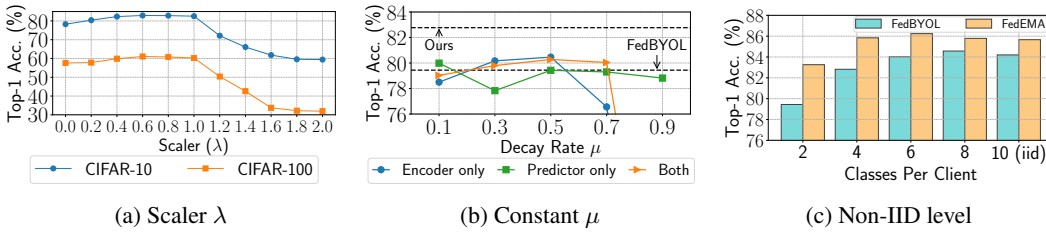

(a) Scaler $\lambda$      (b) Constant $\mu$      (c) Non-IID level

Figure 7: Ablation study on scaler $\lambda$, decay rate $\mu$, and non-IID levels of the CIFAR-10 dataset: (a) analyzes the impact of scaler $\lambda$ on performance; (b) compares using constant $\mu$ on encoder, predictor, or both; (c) studies the impact of different non-IID levels.

sults in the best performance. These results demonstrate the effectiveness of updating both predictor and encoder in FedEMA. More results on other settings are provided in Table 5 in Appendix C.

**Changes of Divergence** Figure 6 illustrates that the divergence between global encoder and online encoder (Eqn 3) decreases gradually as training proceeds. It validates our intuition that more local knowledge is used at the start of training when divergence is larger. Besides, clients can update at their own pace depending on the divergence caused by their local dataset.

**Scaler $\lambda$** We study the impact of $\lambda$ with values in $[0, 2]$ with interval of 0.2 in Figure 7a. $\lambda > 1$ leads to a significant performance drop because it results in $\mu = 1$ at the start of training on CIFAR datasets, implying that no aggregated global network is used. When $\lambda \in (0, 1)$, the performances are consistently better than FedBYOL ($\lambda = 0$) as both local and global knowledge are effectively aggregated. These analyses are mainly suitable for our experiment setting. The range values of $\lambda$ depend on the characteristics of data and the hyper-parameters (e.g., local epoch) of FL settings. A practical way to tune $\lambda$ manually is to understand the divergence by running the algorithm for several rounds and choose the $\lambda$ that scales $\mu$ to (0.5, 1). Nevertheless, we recommend using autoscaler and provide ablation study of $\tau$ of autoscaler in Figure 10a in Appendix C.

**Constant Values of $\mu$** We further demonstrate the necessity of dynamic EMA by comparing with using constant values of $\mu$ in Eqn 1 and 2. Figure 7b shows that a good choice of constant $\mu$ can outperform FedBYOL, but FedEMA outperforms using constant $\mu$ for the online encoder, predictor, or applying both. We also provide results that encoder and predictor use different $\mu$ in Appendix C.

**Non-IID Level** Figure 7c compares the performance of different non-IID levels, ranging from 2 to 10 classes per client on the CIFAR-10 dataset. We use autoscaler for these experiments. FedEMA consistently outperforms FedBYOL in these settings.

## 6 CONCLUSION

We uncover important insights of federated self-supervised learning (FedSSL) from in-depth empirical studies, using a newly introduced generalized FedSSL framework. Inspired by the insights, we propose a new method, Federated Divergence-aware Exponential Moving Average update (FedEMA), to further address the non-IID data challenge. Our experiments and ablations demonstrate that FedEMA outperforms existing methods in a wide range of settings. In the future, we plan to implement FedSSL and FedEMA on larger-scale datasets. We hope that this study will provide useful insights for future research.

## 7 REPRODUCIBILITY STATEMENT

To facilitate reproducibility of experiment results, we first provide basic experimental setups in Section 3.3, including datasets, implementation details, and evaluation protocols. Then, we describe more experimental details in Appendix B, including datasets, data transformation, network architecture, training details, and default settings. Also, we indicate the settings and hyper-parameters of experiments when their settings are different from the default. Moreover, we plan to open-source the codes in the future.

### ACKNOWLEDGMENTS

We would like to thank reviewers of ICLR 2022 for their constructive and helpful feedback. This study is in part supported by the RIE2020 Industry Alignment Fund – Industry Collaboration Projects (IAF-ICP) Funding Initiative, as well as cash and in-kind contribution from the industry partner(s); the National Research Foundation, Singapore under its Energy Programme (EP Award NRF2017EWT-EP003-023) administrated by the Energy Market Authority of Singapore, and its Energy Research Test-Bed and Industry Partnership Funding Initiative, part of the Energy Grid (EG) 2.0 programme, and its Central Gap Fund ("Central Gap" Award No. NRF2020NRF-CG001-027); Singapore MOE under its Tier 1 grant call, Reference number RG96/20.

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

## A    DIFFERENCES OF SELF-SUPERVISED LEARNING METHODS

We study four SSL methods using the FedSSL framework in Section 3. These four SSL methods have two major differences that impact the executions of local training, model communication, and model aggregation. Figure 8 depicts these differences: 1) BYOL and SimSiam have predictors, whereas MoCo and SimCLR do not have them; 2) SimSiam and SimCLR share weights between two encoders, whereas BYOL and MoCo have different parameters for the online and target encoders.

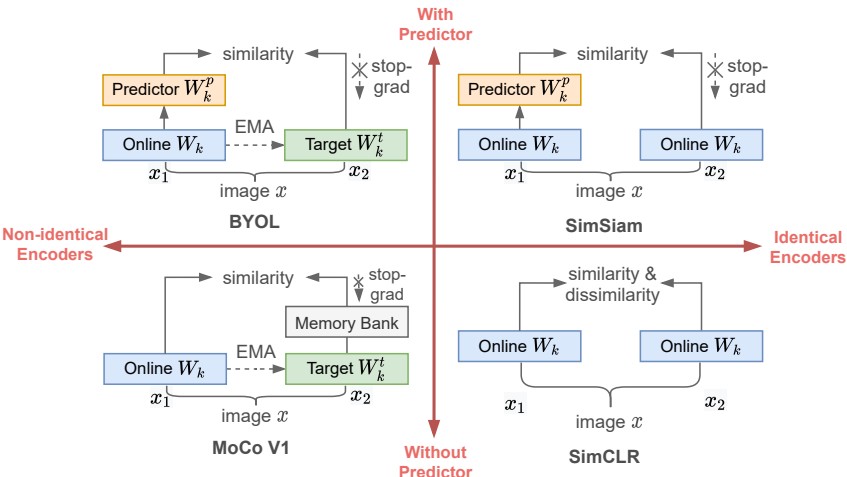

Figure 8: Illustration of differences among four Self-supervised Learning (SSL) methods.

## B    EXPERIMENTAL DETAILS

In this section, we provide more details about the dataset, network architecture, and training and evaluation setups.

### B.1    DATA

**Datasets**   CIFAR-10 and CIFAR-100 are two popular image datasets (Krizhevsky et al., 2009). Both datasets consist of 50,000 training images and 10,000 testing images. CIFAR-10 contains 10 classes, where each class has 5,000 training images and 1,000 testing images. While CIFAR-100 contains 100 classes, where each class has 500 training images and 100 testing images. To simulate federated learning, we equally split the training set into $K$ clients. We simulate non-IID data using label heterogeneity — data among clients is more skewed when each client contains less number of classes. Hence, we simulate different levels of non-IID data with $l$ number of classes per client, where $l = \{2, 4, 6, 8, 10\}$ for CIFAR-10 and $l = \{20, 40, 60, 80, 100\}$ for CIFAR-100. For example, when simulating 5 clients with $l = 4$ classes per client in CIFAR-10, we need $5 \times 4 = 20$ total sets of data over 10 classes. Thus, we split the training images of each class equally into two sets (2,500 images in each set) and assign random four sets without overlapping classes to a client. The setting is IID when each client contains all classes of a dataset. By default, we run experiments with $K = 5$ clients with non-IID setting $l = 2$ classes per client for CIFAR-10 dataset and $l = 20$ classes per client for CIFAR-100 dataset.

**Transformation**   In local training of the FedSSL framework, we take two augmentations of each image as the inputs for online and target networks, respectively. We obtain the augmentations by transforming the images with a set of transformations: For SimCLR, BYOL, SimSiam, and MoCoV2, we adopt the transformations from Chen et al. (2020a); For MoCoV1, we use the transformation described in its paper (He et al., 2020).

Table 5: Top-1 accuracy comparison under linear evaluation protocol on CIFAR datasets. Our proposed FedEMA outperforms all other methods on non-IID settings.

| Method | Architecture | Param. | CIFAR-10 (%) | | CIFAR-100 (%) | |
|---|---|---|---|---|---|---|
| | | | IID | Non-IID | IID | Non-IID |
| Standalone training | ResNet-18 | 11M | 82.42 | 74.95 | 53.88 | 52.37 |
| FedSimCLR | ResNet-18 | 11M | 82.15 | 78.09 | 56.39 | 55.58 |
| FedMoCoV1 | ResNet-18 | 11M | 83.63 | 78.21 | **59.58** | 56.98 |
| FedMoCoV2 | ResNet-18 | 11M | 84.25 | 79.14 | 58.71 | 57.47 |
| FedSimSiam | ResNet-18 | 11M | 81.46 | 76.27 | 49.92 | 48.94 |
| FedBYOL | ResNet-18 | 11M | 84.29 | 79.44 | 54.24 | 57.51 |
| FedU (Zhuang et al., 2021a) | ResNet-18 | 11M | 83.96 | 80.52 | 54.82 | 57.21 |
| FedEMA predictor only (ours) | ResNet-18 | 11M | 84.97 | 81.13 | 55.52 | 57.53 |
| FedEMA encoder only (ours) | ResNet-18 | 11M | 82.88 | 82.39 | 56.06 | 59.74 |
| FedEMA ($\lambda = 0.8$) | ResNet-18 | 11M | **85.59** | **82.77** | 57.86 | **61.21** |
| FedEMA (autoscaler, $\tau = 0.7$) | ResNet-18 | 11M | **86.26** | **83.34** | 58.55 | **61.78** |
| Standalone training | ResNet-50 | 23M | 83.16 | 77.84 | 57.21 | 55.16 |
| FedSimCLR | ResNet-50 | 23M | 82.24 | 80.37 | 57.46 | 56.88 |
| FedMoCoV1 | ResNet-50 | 23M | 87.19 | 82.18 | **64.74** | 59.73 |
| FedMoCoV2 | ResNet-50 | 23M | **87.19** | 79.62 | 63.75 | 59.52 |
| FedSimSiam | ResNet-50 | 23M | 79.64 | 76.7 | 46.28 | 48.8 |
| FedBYOL | ResNet-50 | 23M | 83.90 | 81.33 | 57.75 | 59.53 |
| FedCA (Zhang et al., 2020a) | ResNet-50 | 23M | 71.25 | 68.01 | 43.30 | 42.34 |
| FedU (Zhuang et al., 2021a) | ResNet-50 | 23M | 86.48 | 83.25 | 59.51 | 61.94 |
| FedEMA predictor only (ours) | ResNet-50 | 23M | 83.66 | 81.78 | 57.79 | 60.11 |
| FedEMA encoder only (ours) | ResNet-50 | 23M | 84.66 | 84.91 | 58.52 | 62.51 |
| FedEMA ($\lambda = 0.8$) | ResNet-50 | 23M | 86.12 | **85.29** | 60.96 | **62.53** |
| FedEMA (autoscaler, $\tau = 0.7$) | ResNet-50 | 23M | 85.08 | **84.31** | 59.48 | **62.77** |
| BYOL (Centralized) | ResNet-18 | 11M | 90.46 | - | 65.54 | - |
| BYOL (Centralized) | ResNet-50 | 23M | 91.85 | - | 66.51 | - |

Table 6: Top-1 accuracy comparison on larger numbers of clients with client subsampling: 1) randomly selecting 5 out of 20 clients per round (5/20); 2) randomly selecting 8 out of 80 clients per round (8/80). FedEMA, trained with autoscaler, consistently outperforms FedBYOL in both settings.

| Method | 5/20 clients (%) | | | | 8/80 clients (%) | | | |
|---|---|---|---|---|---|---|---|---|
| | CIFAR-10 | | CIFAR-100 | | CIFAR-10 | | CIFAR-100 | |
| | IID | Non-IID | IID | Non-IID | IID | Non-IID | IID | Non-IID |
| FedBYOL | 83.25 | 74.92 | 49.49 | 47.09 | 73.58 | 63.28 | 41.19 | 41.58 |
| FedEMA (ours) | **84.98** | **75.77** | **55.41** | **52.78** | **73.96** | **64.19** | **41.97** | **43.05** |

## B.2 Network Architecture

**Predictor** The network architecture of the predictor is a two-layer multilayer perceptron (MLP). The two-layer MLP starts from a fully connected layer with 4096 neurons. Followed by one-dimension batch normalization and a ReLU activation function, it ends with another fully connected layer with 2048 neurons.

**Encoder** We use ResNet-18 He et al. (2016) as the default network architecture of the encoder in the majority of experiments. Besides, we also provide results of ResNet-50 in Table 5 and 7.

Table 7: Top-1 accuracy comparison on using 1% and 10% of labeled data for semi-supervised learning on the non-IID settings of CIFAR datasets. FedEMA outperforms all other methods.

| Method | Architecture | Param. | CIFAR-10 (%) | | CIFAR-100 (%) | |
|---|---|---|---|---|---|---|
| | | | 1% | 10% | 1% | 10% |
| Standalone training | ResNet-18 | 11M | 61.37 | 69.06 | 21.37 | 39.99 |
| FedSimCLR | ResNet-18 | 11M | 63.79 | 73.49 | 21.55 | 41.90 |
| FedMoCoV1 | ResNet-18 | 11M | 60.57 | 73.95 | 21.83 | 43.49 |
| FedMoCoV2 | ResNet-18 | 11M | 62.89 | 73.65 | 26.93 | 45.27 |
| FedSimSiam | ResNet-18 | 11M | 67.57 | 74.96 | 25.13 | 41.96 |
| FedBYOL | ResNet-18 | 11M | 70.48 | 76.95 | 30.21 | 47.07 |
| FedU (Zhuang et al., 2021a) | ResNet-18 | 11M | 69.52 | 77.06 | 29.00 | 46.67 |
| FedEMA ($\lambda = 1$) | ResNet-18 | 11M | **72.78** | **79.01** | **32.49** | **49.82** |
| FedEMA (autoscaler, $\tau = 0.7$) | ResNet-18 | 11M | **73.44** | **79.49** | **33.04** | **50.48** |
| Standalone training | ResNet-50 | 23M | 63.65 | 74.30 | 23.18 | 41.43 |
| FedSimCLR | ResNet-50 | 23M | 63.00 | 73.56 | 19.30 | 41.13 |
| FedMoCoV1 | ResNet-50 | 23M | 61.85 | 75.53 | 22.12 | 46.43 |
| FedMoCoV2 | ResNet-50 | 23M | 64.25 | 73.96 | 25.79 | 42.52 |
| FedSimSiam | ResNet-50 | 23M | 61.46 | 15.25 | 16.03 | 29.76 |
| FedBYOL | ResNet-50 | 23M | 69.99 | 76.69 | 26.57 | 45.46 |
| FedCA (Zhang et al., 2020a) | ResNet-50 | 23M | 28.50 | 36.28 | 16.48 | 22.46 |
| FedU (Zhuang et al., 2021a) | ResNet-50 | 23M | 69.76 | 80.25 | 28.42 | 48.42 |
| FedEMA ($\lambda = 1$) | ResNet-50 | 23M | **74.64** | **81.48** | **31.42** | **49.92** |
| FedEMA (autoscaler, $\tau = 0.7$) | ResNet-50 | 23M | **72.52** | **80.68** | **29.68** | **50.75** |
| BYOL (Centralized) | ResNet-18 | 11M | 87.67 | 87.89 | 40.96 | 56.60 |
| BYOL (Centralized) | ResNet-50 | 23M | 89.07 | 89.66 | 41.49 | 60.23 |

Table 8: Top-1 accuracy comparison on various non-IID levels — the number of classes per client on the CIFAR-100 dataset. Update-both means updating both $W_k$ and $W_k^t$ with $W_g$.

| Method | # of classes per client (%) | | | | |
|---|---|---|---|---|---|
| | 2 | 4 | 6 | 8 | 10 (iid) |
| FedBYOL | 57.51 | 56.96 | 55.14 | 54.96 | 54.24 |
| FedSimSiam | 48.94 | 51.08 | 49.05 | 48.09 | 49.92 |
| FedBYOL, update-both | 49.53 | 54.17 | 51.50 | 52.70 | 53.41 |

Our ResNet architecture differs from the implementation in PyTorch (Paszke et al., 2017) in three aspects: 1) We use kernel size $3 \times 3$ for the first convolution layer instead of $7 \times 7$; 2) We use an average pooling layer with kernel size $4 \times 4$ before the last linear layer instead of adaptive average pooling layer; 3) We replace the last linear layer with a two-layer MLP. The network architecture of the MLP is the same as the predictor.

### B.3 TRAINING AND EVALUATION DETAILS

We implement FedSSL in Python using EasyFL (Zhuang et al., 2022), an easy-to-use federated learning platform based on PyTorch (Paszke et al., 2017). The following are the details of training and evaluation.

**Training** We use Stochastic Gradient Descent (SGD) as the optimizer in training. We use $\eta = 0.032$ as the initial learning rate and decay the learning with a cosine annealing (Loshchilov & Hutter, 2017), which is also used in SimSiam. By default, we train $R = 100$ rounds with local

Table 9: Comparison of FedBYOL without exponential moving average (EMA) and stop-gradient (sg) on the CIFAR datasets. FedBYOL w/o EMA and sg can hardly learn, but updating both $W_k$ and $W_k^t$ with $W_g$ (update-both) enables it to achieve comparable results.

| Method | CIFAR-10 (%) | | CIFAR-100 (%) | |
|---|---|---|---|---|
| | IID | Non-IID | IID | Non-IID |
| FedBYOL w/o EMA | 54.11 | 50.20 | 23.82 | 25.83 |
| FedBYOL w/o EMA and stop-grad | 21.21 | 11.97 | 3.74 | 2.79 |
| FedBYOL w/o EMA and stop-grad, update-both | 82.29 | 68.75 | 48.74 | 41.91 |
| FedBYOL | **84.29** | **79.44** | **54.24** | **57.51** |

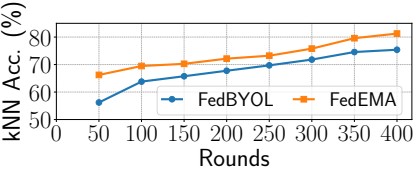

| Method | Rounds $R$ (%) | | | |
|---|---|---|---|---|
| | 100 | 200 | 300 | 400 |
| FedBYOL | 79.08 | 82.23 | 83.77 | 86.09 |
| FedEMA (ours) | **80.78** | **82.41** | **84.08** | **86.51** |

Figure 9: Comparison of FedBYOL and FedEMA on various total training rounds $R$ on the non-IID setting of the CIFAR-10 dataset. FedEMA consistently outperforms FedBYOL.

epochs $E = 5$ and batch size $B = 128$ using $K = 5$ clients. We simulate training of $K$ clients on $K$ NVIDIA V100 GPUs and employ the PyTorch Paszke et al. (2017) communication backend (NCCL) for communications between clients and the server. If not specified, we use $\lambda = 1$ by default or autoscaler with $\tau = 0.7$ for FedEMA. As for experiments of FedU, we follow the hyper-parameters described in paper (Zhuang et al., 2021a).

**Cross-silo FL vs Cross-device FL** This paper primarily focuses on cross-silo FL where clients are stateful with high availability. Clients can cache local models and carry these local states from round to round. Extensive experiments demonstrate that FedEMA achieves the best performance under this setting. On the other hand, cross-device FL assumes there are millions of stateless clients that might participate in training just once. Due to the constraints of experimental settings, the majority of studies conduct experiments with at most hundreds of clients (Wang et al., 2020; Jeong et al., 2021). FedEMA can work under such experimental settings by caching the states of clients in the server. When the number of clients scales to millions, FedEMA degrades to FedBYOL that updates both encoders — without keeping any local states.

**Evalaution** We assess the quality of learned representations using linear evaluation (Kolesnikov et al., 2019; Grill et al., 2020) and semi-supervised learning (Zhai et al., 2019; Chen et al., 2020a) protocols. We first obtain a trained encoder (or learned representations) using full training set for linear evaluation and 99% or 90% of the training set for semi-supervised learning (excluding the 1% or 10% for fine-tuning). Then, we conduct evaluations based on the trained encoder. For linear evaluation, we train a new fully connected layer on top of the frozen trained encoder (fixed parameters) for 200 epochs, using batch size 512 and Adam optimizer with learning rate 3e-3. For semi-supervised learning, we add a new two-layer MLP on top of the trained encoder and fine-tune the whole model using 1% or 10% of data for 100 epochs, using batch size 128 and Adam optimizer with learning rate 1e-3. In both evaluation protocols, we remove the two-layer MLP of the encoder by replacing it with an identity function.

## C ADDITIONAL EXPERIMENTAL RESULTS AND ANALYSIS

In this section, we provide more experimental results of algorithm comparisons and further analyze FedEMA in different data amounts, training rounds $R$, and batch sizes $B$.

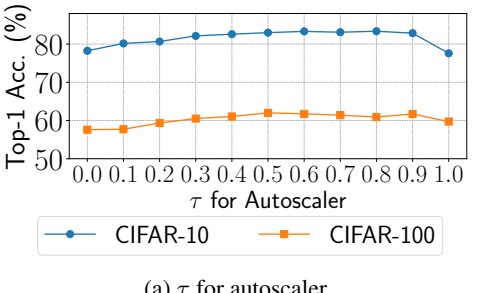
(a) $\tau$ for autoscaler

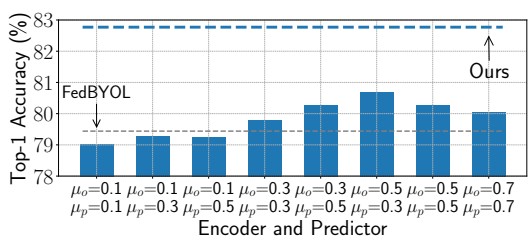
(b) Combinations of $\mu_o$ and $\mu_p$

Figure 10: Ablation study on $\tau$ for autoscaler and combinations of constant $\mu$: (a) analyzes the impact of $\tau$ on performances; 2) presents top-1 accuracy of using different combinations of constant $\mu_o$ on the online encoder and constant $\mu_p$ on the predictor.

Table 10: Top-1 accuracy comparison on various batch sizes $B$ on the non-IID setting of CIFAR-10 dataset. The batch size should not be either too small or too large. Besides, FedEMA outperforms FedBYOL.

| Method | Batch Sizes $B$ (%) | | | | | |
|---|---|---|---|---|---|---|
| | 16 | 32 | 64 | 128 | 256 | 512 |
| FedBYOL | 68.74 | 72.90 | 78.58 | 79.44 | 79.80 | 77.74 |
| FedEMA (ours) | **74.03** | **79.06** | **82.18** | **83.34** | **82.19** | **80.51** |

## C.1 MORE EXPERIMENTAL RESULTS

Table 5 presents top-1 accuracy comparison under linear evaluation of a wide range of methods on CIFAR datasets using both ResNet-18 and ResNet-50. It supplements the algorithm comparisons in Section 3.4 and 5.1. Interestingly, FedMoCoV1 achieves good performances on IID settings of CIFAR-100 dataset. Since decentralized data are mostly non-IID, we focus more on the non-IID setting. FedEMA outperforms all the other methods in non-IID settings of CIFAR datasets. We use $\lambda = 0.8$ when using ResNet-18 and $\lambda = 1$ when using ResNet-50.

Table 6 shows results of scaling to larger numbers of clients $K$ with subsampling clients in each training round. We run two sets of experiments: 1) randomly selecting 5 out of 20 clients in each round with local epoch $E = 5$ and total rounds $R = 400$; 2) randomly selecting 8 out of 80 clients in each round with local epoch $E = 2$ and total rounds $R = 800$. We run FedEMA with autoscaler. FedEMA consistently outperforms FedBYOL with both encoders updated. We conduct these experiments using ResNet-18.

Table 7 supplements the semi-supervised learning results on Table 4, providing additional results using ResNet-50 as the network architecture for the encoder. FedEMA consistently outperforms all the other methods.

Besides, Table 8 and 9 compare FedSimSiam, FedBYOL, and variances of FedBYOL to further demonstrate the insights from empirical studies. They supplement results in Table 2 and Figure 3.

## C.2 FURTHER ANALYSIS

**$\tau$ for Autoscaler** We analyze the impact of $\tau$ on performances in Figure 10a. Generally, using autoscaler with $\tau \in (0, 1)$ is better than FedBYOL ($\tau = 0$). The performance of $\tau = 1$ yields worse results because only local knowledge are used in model update (the global knowledge is neglected) as discussed in Section 4. Besides, performances of $\tau \in [0.5, 1)$ are generally better other values, which verifies our intuition discussed in Section 4. These results also show that we can achieve even higher performance on the CIFAR-100 dataset on Table 3 by tunning $\tau$. We run experiments on non-IID settings using ResNet-18.

Table 11: Comparison of needed communication rounds to reach target accuracy using different local epochs $E$ on the non-IID setting of the CIFAR-10 dataset. $E = 1$ is unable to reach 80% in 100 rounds. A larger $E$ can reduce communication costs by increasing the computation cost.

| Target accuracy | Communiation (rounds) | | | | Computation (epochs) | | | |
|---|---|---|---|---|---|---|---|---|
| | $E = 1$ | $E = 5$ | $E = 10$ | $E = 20$ | $E = 1$ | $E = 5$ | $E = 10$ | $E = 20$ |
| 70% | 90 | 40 | 10 | 8 | 90 | 200 | 100 | 160 |
| 80% | - | 80 | 50 | 40 | - | 400 | 500 | 800 |

Table 12: Top-1 accuracy comparison of various data amounts in clients and different numbers of clients. Increasing the number of clients does not improve performance, whereas increasing the data amount of clients results in better performance.

| # of clients | K = 5 | | | | K = 10 | K = 20 |
|---|---|---|---|---|---|---|
| Data amount | 10% | 25% | 50% | 100% | 50% | 25% |
| FedBYOL | 43.27 | 65.14 | 76.11 | 78.25 | 75.10 | 63.95 |
| FedEMA (ours) | 44.33 | 67.46 | 79.49 | 82.54 | 79.20 | 66.61 |

**Constant $\mu$** To further illustrate the effectiveness of our dynamic EMA, we provide results of using different combinations of constant $\mu_o$ on the online encoder and constant $\mu_p$ on the predictor in Figure 10b. The results of $\mu_o = 0.9$ and $\mu_p = 0.9$ is only 54.52%, which is far lower than the others. Among these combinations, $\mu_o = 0.5$ and $\mu_p = 0.3$ achieve the best performance. It suggests that better performances may be achieved if we can construct different dynamic $\mu$ for the encoder and the predictor, while we leave this interesting insight for future exploration. Although good choices of constant $\mu_o$ and $\mu_p$ achieve better performance than FedBYOL, FedEMA consistently outperforms all these methods. These results complement Figure 7b in the main manuscript.

**Impact of Training Rounds $R$** Figure 9 compares FedBYOL and FedEMA with increasing number of training (communication) rounds $R$. Performances of both FedBYOL and FedEMA increases as training proceeds and FedEMA consistently outperforms FedBYOL. We run these experiments with $\lambda = 0.5$ for FedEMA on the non-IID setting of CIFAR-10 dataset.

**Impact of Batch Size $B$** We investigate the impact of batch size in Table 10. The performances of batch size $B = 128$ and $B = 256$ are similar, outperforming the other batch sizes. It indicates that the batch size should not be either too small or too large. Besides, FedEMA outperforms FedBYOL in all batch sizes. We run the experiments with autoscaler ($\tau = 0.7$) on the non-IID setting of the CIFAR-10 dataset.

**Communication vs Computation Cost** Table 11 shows the needed communication rounds and computation epochs to reach a target accuracy using different local epochs $E$ with FedEMA. Increasing $E$ reduces communication cost as the needed rounds decrease, but it generally requires a higher computation cost. For example, compared with $E = 5$ that needs 80 rounds to reach 80% with 400 epochs of computation, $E = 20$ only uses 40 rounds but needs 800 epochs computation cost. These results indicate the trade-off between communication cost and computation cost.

**Data Amount** Table 12 shows that increasing the data amount improves the performance significantly. By default, we split the CIFAR-10 dataset into 5 clients, each client contains 10,000 training images, denoting as 100% data amount. As a result, $p\%$ data amount means that each client contains $10,000 * p\%$ images. For example, 25% data amount means that each client contains 2,500 images. With lesser data points in each client, we can construct more clients to conduct training as the total data amount is fixed. Table 12 shows that when the data amount is same in clients, increasing the number of clients in each training round do not improve performance. However, increasing the data amount in each client increases the performance significantly. These results indicate that it is important for clients to have sufficient data to participate in training in FedSSL.

