# OpenReview forum: "Divergence-aware Federated Self-Supervised Learning"
_ICLR.cc/2022/Conference — ICLR 2022 Poster_

### Official Review · Reviewer_AcAS · 2021-10-30

**Correctness:** 3
**Technical Novelty And Significance:** 2
**Empirical Novelty And Significance:** 2
**Recommendation:** 5
**Confidence:** 2

**Main Review:**

I think overall, the idea to solve the non-IID problem in Federated learning is good. However, it is not immediately clear to me why the solution works from a theortical perspective. The authors have demonstrated that the approach is superior by a quite significant margin emperically on CIFAR-10 and CIFAR-100.

(1) The superscript and subscript of W have been defined. But W itself has not been defined in the paper. From the context, I assume these are the parameters of your model.

(2) How do you select \lambda?

**Summary Of The Paper:**

The authors propose a new approach called Federated Divergence-aware Exponential moving Average update (FedEMA) to avoid the IID assumption. FedEMA is built onto of FedSSL which is a framework for self-supervised learning in a Federated Learning context. The authors proposes a new approach to fuse the local and global knowledge effectively through a EMA update, where the decay rate of EMA is dynamically measured by a divergence.


**Summary Of The Review:**

Overall an interesting idea. However, I thought the novelty proposed is a bit incremental and should have been more thoroughly investigated. For example more experiments could have been run and it would be good to theoretically show why this works. Although from Section 3.5, I can understand the intuition behind why this works. Overall, I felt that the contribution that this paper proposes does not meet the requirement to be published at ICLR.

---

> ### Author Response · Authors · 2021-11-20
> **Reply to Reviewer AcAS**
>
> Thank you for your feedback. We address your detailed comments below:
>
> **Q1**: The superscript and subscript of W have been defined. But W itself has not been defined in the paper. From the context, I assume these are the parameters of your model.
>
> **A1**: Yes. W itself stands for the parameters of the model. We have revised it in Section 3.2 (last line) to prevent misunderstandings.
>
> **Q2**: How do you select $\lambda$? (Same as Q1 from Reviewer wF5G)
>
> **A2.1**: Thank you for raising this important question. Inspired by your question, we further propose a practical \textit{autoscaler} to calculate a personalized $\lambda_k$ for each client $k$ automatically. The formula is $\lambda_k = \frac{\tau}{||W_g^{r+1} - W_k^r||}$, where $\tau \in [0, 1]$ is the expected value of $\mu$ at round $r$. We calculate $\lambda_k$ only once at the earliest round $r$ that client $k$ is sampled for training. When the same set of clients are sampled for training, $\lambda_k = \frac{\tau}{||W_g^1 - W_k^0||}$ is calculated at round $r=1$. Since model divergence is larger at the start of training (Figure 6), it is practical to choose larger $\tau \in [0.5, 1)$; $\tau = 1$ is not considered because only local knowledge is used when $\tau = 1$. We have provided ablation studies on $\tau$ in Figure 10(a) in Appendix C. Autoscaler achieves comparable, sometimes better results, than choosing $\lambda$ manually, as compared in the folloing three tables. We use $\tau = 0.7$ by default in experiments.
>
> Table 1: Top-1 accuracy comparison on linear evaluation protocol.
>
> | Methods | CIFAR-10 IID | CIFAR-10 Non-IID | CIFAR-100 IID | CIFAR-100 Non-IID |
> | --- | --- | --- | --- | --- |
> | FedEMA ($\lambda=0.8$) | 85.59 $\pm$ 0.25 | 82.77 $\pm$ 0.08 | 57.86 $\pm$ 0.15 | 61.21 $\pm$ 0.54 |
> | FedEMA (autoscaler) | 86.26 $\pm$ 0.26 | 83.34 $\pm$ 0.39 | 58.55 $\pm$ 0.34 | 61.78 $\pm$ 0.14 |
>
> Table 2: Top-1 accuracy comparison on semi-supervised learning evaluation.
>
> | Methods | CIFAR-10 IID | CIFAR-10 Non-IID | CIFAR-100 IID | CIFAR-100 Non-IID |
> | --- | --- | --- | --- | --- |
> | FedEMA ($\lambda=1$) | 72.78 $\pm$ 0.66 | 79.01 $\pm$ 0.30 | 32.49 $\pm$ 0.22 | 49.82 $\pm$ 0.36 |
> | FedEMA (autoscaler) | 73.44 $\pm$ 0.22 | 79.49 $\pm$ 0.34 | 33.04 $\pm$ 0.23 | 50.48 $\pm$ 0.11 |
>
> Table 3: Top-1 accurarcy comparison on various batch sizes on the non-IID setting of CIFAR-10 dataset. We use λ = 0.4 for B = 32 and λ = 0.6 for B = 64.
>
> | Methods | B = 32 | B = 64 | B = 128 |
> | --- | --- | --- | --- |
> | FedEMA (manual $\lambda$) | 79.55 | 81.31 | 82.77 | 82.70 | 79.93 |
> | FedEMA (autoscaler) | 79.06 | 82.18 | 83.34 |
>
> **A2.2**: A practical way to tune $\lambda$ manually is to understand the divergence by running the algorithm for several rounds and choose the $\lambda$ that scales $\mu$ to (0.5, 1). Nevertheless, we recommend using autoscaler. We have also revised the discussion on Scaler in Section 5.2 in the manuscript to reflect these comments to prevent misunderstanding.
>
> **Q3**: Overall an interesting idea. However, I thought the novelty proposed is a bit incremental and should have been more thoroughly investigated. For example more experiments could have been run and it would be good to theoretically show why this works.
>
> **A3.1**: Thank you for raising the concern. We would like to clarify that learning representations from unlabeled decentralized data while preserving data privacy is still a nascent field. Previous works only focus on single self-supervised learning (SSL) methods. While SSL methods are evolving rapidly and new methods (like SimSiam [2], MoCoV2, SimCLR) are emerging, it is crucial to understand the fundamental components of FedSSL. In this paper, we first introduce a generalized FedSSL framework that embraces existing SSL methods and has flexibility catering to future methods. Then, we took an investigative approach to deeply investigate the fundamental components of FedSSL to build up intuitions and practical guidelines for the generic FedSSL framework. Based on these insights, we propose a new model update approach, FedEMA, which improves the performance substantially.
>
> **A3.2**: The intuition of FedEMA is to retain more local knowledge when divergence is large and incorporate more global knowledge when divergence is small. It can also be understood in a way of smoothing the integration of local and global knowledge. Due to time constraints, we will consider theoretical proofs for the algorithm and the framework in our future work. On the other hand, we have conducted and presented results of more than 200 experiments in the manuscript to cover a wide range of settings; we would appreciate it if the reviewer could provide us with more details on what are the expected experiments to run. We are very willing to run them and integrate them into the paper.

---

> ### Author Response · Authors · 2021-11-24
> **Need further clarification?**
>
> Thanks for your comments and questions on our work. Is there still any unclear point about the paper and the rebuttal? Although we cannot update the rebuttal revision, do you have suggested experiments that we could/should further provide here?

---

### Official Review · Reviewer_Nu54 · 2021-11-01

**Correctness:** 4
**Technical Novelty And Significance:** 3
**Empirical Novelty And Significance:** 3
**Recommendation:** 8
**Confidence:** 4

**Main Review:**

The paper follows a clear red line and is easy to follow. The empirical insights are interesting and mostly well motivated and explained. I have a few open question related to the paper and some suggestions which would make this a very good paper.

The most successful methods, as well as FedEMA rely on the availablitiy of a local target network. I.e. the clients are state-full between rounds. In the general (sightly more mature) supervised FL literature, this setting is called the 'cross-silo' setting as opposed to the cross-device setting, in which we assume the federation to consist of millions of devices, each of which might take part in the federation just once. This distinction is missing in this paper and I encourage the authors to make this more explicit.
Related to this comment is my biggest issue with the experimental claims in this paper. For the federated setting $K=5$ clients is not enough. Even for a paper targeting cross-silo FL, I would suggest to experiment with more clients to make the insights more valuable. Table 11 considers up to $K=20$ clients, however the authors also reduce the amount of data per-client (in addition to spreading the available data over a larger amount of clients, as far as I understand).
Along the same lines, FL usually does not assume full client availability, i.e. not all $K$ clients are available for updates at every round. Subsampling clients is especially interesting in the context of state-full clients, since their local models might become stale. Also as the amount of data locally becomes smaller, methods that rely on a local target encoder might overfit, something that FedEMA might remedy. I would encourage the authors to explore that angle or clearly position their work in terms of which niche of the heterogeneous  FL space they target.
How representative of SSL methods are small-scale data-sets such as cifar10/cifar100? Maybe the authors could compare their scale to the literature. I encourage them to evaluate some experimental insights on larger data-sets such as Imagenet.

Server-side optimization benefits from more advanced update rules than simply averaging client updates. E.g. https://arxiv.org/abs/2003.00295 explore this design space. Is there a reason the authors choose to do averaging only? Otherwise I'd encourage the authors to explore the influence of these optimisers.

Your discussion of the scaler $\lambda$ mentions that setting $\lambda>1$ leads to $\mu=1$. I figure that would depend a lot on the federation characteristics that determine the divergence between local and global model over time. A such, it is a very general statement. Maybe you could track the value of $\mu$ over time for different settings of $\lambda$ for your experiments.

Experimental setting:
- What hyper-parameter ranges have the authors explored for their experiments?
- Is centralised BYOL done with same hyperparameters? E.g. can you do centralised performance with small batch-sizes?
- Is linear evaluation done on the locally fine-tuned models locally or on the most recent global model with data at the server?

Misc:
- What is the difference between the pink line/model in Figure 3 and the last row in Table 2?
- What time is indexed by $t$ in 3.2? Is that local time-steps per client or the global update steps? I.e. is the target encoder updated every mini-batch or every global round only? How is this different from equation 1 and 2? Are you describing 2-level updates? Algorithm 1 suggests the answer, but I encourage to make this more clear in the text. I also suggest adding superscript $0$ to lines 13/14 in Alg 1.

Disclaimer: I am not an expert in SSL methods, which is why I appreciated Appendix A and corresponding discussions in the main text.


**Summary Of The Paper:**

The authors introduce an abstraction of self-supervised learning (SSL) algorithms in the federated setting, which they term FedSSL. They empirically evaluate several existing SSL methods in the federated setting and derive a new algorithm, FedEMA based on their insights.

**Summary Of The Review:**

This paper provides an interesting insight into self-supervised training in FL. It motivates its proposed method by empirical insights on cifar10(0). My major issue with the paper is that its empirical scale is quite small in terms of what makes the FL scenario challenging. On the other hand, they have very broad empirical analyses.
If the authors extend their experimental evaluation along these scale-related dimensions and clarify their positioning within the FL literature, I will consider raise my score.

=== Post-Rebuttal ===
The authors have addressed my concerns sufficiently and I have updated my score accordingly.

---

> ### Author Response · Authors · 2021-11-20
> **Reply to Reviewer Nu54 (1/2)**
>
>
> We thank the reviewer for valuable comments and suggestions that have greatly improved the paper.
>
> **Q1**: Clearly position the work and discussion of cross-silo FL vs cross-device FL.
>
> **A1**: We have followed your suggestion and revised this paper to primarily focus on the cross-silo FL where clients are stateful with high availability (fourth paragraph in the introduction). Besides, we also discuss cross-silo FL and cross-device FL in Appendix B.3.
>
> **Q2**: Subsampling; Scalability; Small data amoumt.
>
> **A2**: We have revised Algorithm 1 to support random client sampling on every training round. Since local models of clients might become stale, when they are newly sampled, we update them using the global model; when they are sampled in the previous training round, we apply EMA to update their local models. We have followed your suggestion to further increase the scale of evaluation to K = 80 clients. We have added two experiments for subsampling with a larger amount of clients: 1) randomly sampling 5 out of 20 clients per round; 2) randomly sampling 8 out of 80 clients per round. Table 1 and 2 shows that FedEMA using autoscaler outperforms FedBYOL and FedBYOL with updating both encoders. When scaling up the number of clients, we can only reduce the data amount per client due to the size of the dataset. We are interested in further improving the performance under small data amounts (625 samples per client when K = 80 clients), while we think it is a bit out of the scope of this paper as comparisons of different algorithms are fair if they are under the same setting. We will consider it as our future work.
>
> Table 1: Top-1 accuracy comparison on client sampling: randomly sampling *5 out of 20* clients per round.
>
> | Methods | CIFAR-10 IID | CIFAR-10 Non-IID | CIFAR-100 IID | CIFAR-100 Non-IID |
> | --- | --- | --- | --- | --- |
> | FedBYOL | 83.25 | 74.92 | 49.49 | 47.09 |
> | FedEMA | 84.98 | 75.77 | 55.41 | 52.78 |
>
> Table 2: Top-1 accuracy comparison on client sampling: randomly sampling *8 out of 80* clients per round.
>
> | Methods | CIFAR-10 IID | CIFAR-10 Non-IID | CIFAR-100 IID | CIFAR-100 Non-IID |
> | --- | --- | --- | --- | --- |
> | FedBYOL | 73.51 | 63.33 | 41.70 | 41.59 |
> | FedEMA | 73.76 | 63.86 | 42.59 | 42.95 |
>
> **Q3**: How representative of SSL methods are small-scale data-sets such as cifar10/cifar100? Encourage to evaluate on larger data-sets such as Imagenet.
>
> **A3**: We have implemented the training in Imagenet, but due to time and resource constraints, it is challenging for us to obtain the results. On the other hand, CIFAR10/100 are one of the most commonly used datasets in FL [1][2]. We follow our previous works [3][4] to conduct experiments on these datasets. We believe that important insights discovered from these datasets are helpful for future research on larger datasets and we will continue our investigation.
>
> **Q4**: Consider server-side optimization E.g. https://arxiv.org/abs/2003.00295.
>
> **A4**: Thank you for raising this interesting point. Since federated averaging (FedAvg) is the most widely adopted algorithm for model aggregation, we use it by default. As suggested, we implement FedYogi (the server-side optimization that achieves the best performance in the mentioned paper) and present the results on the non-IID setting of CIFAR-10 below. Replacing simple averaging with FedYogi does not lead to better performance. We agree that some server-side optimizations could further boost the performance, but it would require further careful considerations, which we will consider in our future works.
>
> Table 3: Top-1 accuracy comparison with and without FedYogi optimization in the server. Experiments are run on the non-IID CIFAR-10 dataset using ResNet-18.
>
> | Methods | with FedYogi | without FedYogi |
> | --- | --- | --- |
> | FedBYOL | 34.01 | 79.44 |
> | FedEMA | 34.04 | 82.77 |
>
> **Q5**: Your discussion, the scaler λ mentions that setting λ>1 leads to μ=1, is too general.
>
> **A5**: Thank you for raising the concern. You are right that the relation between λ and μ depends on federation characteristics. The value of $\mu$ is tacked in Figure 6 and the discussion (λ>1 leads to μ=1) is discovered empirically from our experiments; we have revised the manuscript to make it clear that this argument is specific to the non-IID settings of CIFAR datasets. Relating to this comment, we have further provided practical guidelines in choosing $\lambda$ in the discussion and propose a new *autoscaler* method to calculate a personalized $\lambda$ automatically for each client.

---

> > ### Author Response · Authors · 2021-11-20
> > **Reply to Reviewer Nu54 (2/2)**
> >
> > Experimental setting:
> >
> > **Q6**: What hyper-parameter ranges have the authors explored for their experiments?
> >
> > **A6**: We explored the following hyper-parameter ranges: batch size B = {32, 64, 128, 256}, local epoch E = {1, 5, 10, 20}, number of clients K = {5, 10, 20}, training rounds R = {100, 200, 300, 400}, learning rate $\eta = 0.032$, $λ \in [0, 2]$. By default, we use B = 128, E = 5, K = 5, and R = 100.
> >
> > **Q7**: Is centralised BYOL done with same hyperparameters? E.g. can you do centralised performance with small batch-sizes?
> >
> > **A7**: Centralized BYOL is trained with the same hyperparameters (batch size and learning rate). To standardize the total computation, we train centralized BYOL with 500 epochs (calculated by multiplying 100 rounds with 5 local epochs). Table 5 below presents the results of centralized training with smaller batch sizes. By adjusting the learning rate proportionally as changes in the batch size, centralized training can achieve results comparable to B = 128 with smaller batch sizes.
> >
> > Table 5: Top-1 accuracy comparison of centralized BYOL on different batch sizes on the non-IID setting of CIFAR datasets.
> >
> > | Batch Size | Learning Rate | CIFAR-10 | CIFAR-100 |
> > | --- | --- | --- | --- |
> > | 128 | 0.032 | 90.46% | 65.54% |
> > | 32 | 0.008 | 89.46% | 63.15% |
> > | 16 | 0.004 | 88.01% | 60.60% |
> > | 32 | 0.032 | 80.83% | 42.45% |
> >
> > **Q8**: Is linear evaluation done on the locally fine-tuned models locally or on the most recent global model with data at the server?
> >
> > **A8**: We conduct evaluations using the most recent global model with data at the server.
> >
> > Misc:
> >
> > **Q9**: What is the difference between the pink line/model in Figure 3 and the last row in Table 2?
> >
> > **A9**: They are different in the architecture for local training. The network architecture of the last row in Table 2 (N1: FedBYOL with both EMA and sg, update-both) is the standard BYOL, while the pink line in Figure 3 (N2: FedBYOL w/o EMA and sg, update-both) removes the EMA and stop-gradient (sg) components of BYOL. As a result, in local training, the target encoder of N1 is updated by EMA of the online encoder, while the target encoder of N2 is updated by gradient-descent.
> >
> > **Q10**: What time is indexed by t in 3.2? Encourage to make this more clear in the text. I also suggest adding superscript 0 to lines 13/14 in Alg 1.
> >
> > **A10**:  In Section 3.2, we use $W_k^t$ to denote the target encoder of client $k$. Target encoder is firstly updated every mini-batch by moving average from the online encoder in local training. We have revised it to prevent misunderstanding. We have also revised Algorithm 1 to consider the situation when r = 0 and clarify that the online network, $W_k^o = (W_k, W_k^p)$, is the concatenation of the online encoder $W_k$ and the predictor $W_k^p$ in Figure 4 (besides Algorithm 1).
> >
> > References:
> >
> > [1] Federated Learning with Matched Averaging. ICLR 2020.
> >
> > [2] HeteroFL: Computation and communication efficient federated learning for heterogeneous clients. ICLR 2021.
> >
> > [3] Collaborative unsupervised visual representation learning from decentralized data. ICCV 2021.
> >
> > [4] Federated Unsupervised Representation Learning. ArXiv 2020.

---

> > > ### Comment · Reviewer_Nu54 · 2021-11-23
> > > **Thank you for the rebuttal**
> > >
> > > I thank the authors for their extensive revision and reaction to my and the other reviewer's comments.
> > > Most of my questions/comments have been addressed to my satisfaction.
> > > I am very surprised about the absolutely detrimental performance of FedYogi, maybe the authors can try a more sensible hyper-parameter choice in a future update.
> > > I believe that with the focus on the cross-silo setting, this paper is properly targeted. with the modifications to Alg1, FedEMA gracefully degenerates into FedBYOL as the number of clients becomes larger and the local state becomes stale.
> > > As a minor concern about Alg 1 I find the 'if not \lambda_k' notation confusing. In Line 6, the authors provide \lambda_k to the client without properly initialising/defining it.
> > >
> > > I am almost ready to raise my score to 8, the only thing I would like to have the author's insights on are:
> > > - How does the model perform if instead of overwriting the stale local model, the authors use it for the update in Eq 17/18?
> > > - Reviewer47jX raises the point that the (almost same) baseline in the referenced arxiv work has much better performance than your implementation. The same paper seems to be submitted concurrently to ICLR (and have issues, e.g. Figure 5a is strange...). Regardless, could you comment on the differences that lead to these differences? with alpha=0.1 and dirichlet sampling, the data-distribution is roughly between 2-4 unique labels per client.

---

> > > > ### Author Response · Authors · 2021-11-24
> > > > **Thank you for further feedback**
> > > >
> > > > Thank you for your further feedback. We are glad that most of your concerns are addressed. Below, we respond with more details to your comments.
> > > >
> > > > **Q1**: I am very surprised about the absolutely detrimental performance of FedYogi, maybe the authors can try a more sensible hyper-parameter choice in a future update.
> > > >
> > > > **A1**: We are also curious to understand how to further improve performance with server optimization and will report results if we have new findings.
> > > >
> > > > **Q2**: As a minor concern about Alg 1 I find the 'if not \lambda_k' notation confusing. In Line 6, the authors provide \lambda_k to the client without properly initializing/defining it.
> > > >
> > > > **A2**: Thank you for raising the concern. We have followed your suggestion and revised the paper: 1) initialize $\lambda_k$ to be *null* for all clients; 2) change "*not* $\lambda_k$" to "$\lambda_k$ *is null*". (updated before deadline)
> > > >
> > > > **Q3**: How does the model perform if instead of overwriting the stale local model, the authors use it for the update in Eq 17/18?
> > > >
> > > > **A3**: We provide the results of always using the stale model to update (in the first row) in Table 1 and 2 below. It underperforms FedBYOL that *update both encoders* and FedEMA that only use the stale model from *previous round* (Algorithm 1). It is because the stale model becomes too outdated to contribute to the latest training in random client sampling. For example, client A is firstly selected in round $r =0$ and then selected in round $r = 10$; using the outdated model from $r = 0$ to update latest training in $r = 10$ would have negative impact in training. The frequency of sampling the same client is correlated to the fraction of sampled clients (25\% in Table 1 and 10\% in Table 2): the negative impact of the stale model is amplified from 5/20 clients to 8/80 clients. On the other hand, the performance is boosted if we only use the local model from the previous round (the third row in Table 1 and 2). Such improvement degrades as the decrease of the fraction of sampling from Table 1 and Table 2. And as discussed in [previous reply](https://openreview.net/forum?id=oVE1z8NlNe&noteId=Xsi3IwHveoa), FedEMA degrades gracefully to FedBYOL that update both encoders when extending to large-scale cross-device FL.
> > > >
> > > > By analyzing these results, we suspect that the degree of staleness (how stale the model is) could worth consideration. We use the local model only from the previous round, while it could be good to use the model from the previous few rounds. The degree of staleness to consider may be related to the stage of training: keeping the stale models for fewer rounds in the beginning stage of training and keeping them for more rounds in the later stage of training when the training is converging. These link to your previous comment *"Subsampling clients is especially interesting in the context of state-full clients, since their local models might become stale"*. Designing other client sampling strategies could further boost the performance of FedEMA. We feel there is room for further investigation and would like to open such possibilities for future research.
> > > >
> > > >
> > > > Table 1: Top-1 accuracy comparison on client sampling: randomly sampling 5 out of 20 clients per round (5/20).
> > > >
> > > > | Methods | CIFAR-10 IID | CIFAR-10 Non-IID | CIFAR-100 IID | CIFAR-100 Non-IID |
> > > > | --- | --- | --- | --- | --- |
> > > > | FedEMA, always using stale model | 64.87 | 55.59 | 29.78 | 30.91 |
> > > > | FedBYOL | 83.25 | 74.92 | 49.49 | 47.09 |
> > > > | FedEMA | 84.98 | 75.77 | 55.41 | 52.78 |
> > > >
> > > > Table 2: Top-1 accuracy comparison on client sampling: randomly sampling 8 out of 80 clients per round (8/80).
> > > >
> > > > | Methods | CIFAR-10 IID | CIFAR-10 Non-IID | CIFAR-100 IID | CIFAR-100 Non-IID |
> > > > | --- | --- | --- | --- | --- |
> > > > | FedEMA, always using stale model | 37.69 | 38.60 | 15.98 | 12.36 |
> > > > | FedBYOL | 73.51 | 63.33 | 41.70 | 41.59 |
> > > > | FedEMA | 73.76 | 63.86 | 42.59 | 42.95 |
> > > >
> > > > **Q4**: Reviewer47jX raises the point that the (almost same) baseline in the referenced arxiv work has much better performance than your implementation. The same paper seems to be submitted concurrently to ICLR (and have issues, e.g. Figure 5a is strange...). Regardless, could you comment on the differences that lead to these differences? with alpha=0.1 and dirichlet sampling, the data-distribution is roughly between 2-4 unique labels per client.
> > > >
> > > > **A4**: Thank you for raising this point. The differences in hyper-parameters and experiment settings lead to the differences in results; we can achieve similar results using similar hyper-parameters. We have provided detailed comments in the [reply](https://openreview.net/forum?id=oVE1z8NlNe&noteId=xG7bKiFPI0K) to reviewer 48jX.

---

### Official Review · Reviewer_wF5G · 2021-11-02

**Correctness:** 3
**Technical Novelty And Significance:** 2
**Empirical Novelty And Significance:** 3
**Recommendation:** 6
**Confidence:** 3

**Main Review:**

NOVELTY & SIGNIFICANCE

The key innovation of this paper is the specific formulae -- Eq. (3) -- that computes the decay rate in the EMA recipe which is driven by a scaler and the normalized L2-distance between the local & global online net. While this appears minimal in terms of algorithmic novelty, the empirical validation is quite extensive and shows strong improvement over existing baselines.

Thus, despite the simple technical innovation, I have a positive opinion of this paper. In fact, I think this is a pretty well-structured investigative process that arrives at a fine-tuned version of federated SSL with improved performance. It also re-affirms or discovers relatively useful empirical practices that boosts the performance of federated SSL, particularly in highly non-IID settings.

On another note, however, it seems the performance would depend on whether we have a correct choice of lambda. For Tables 3-4, how would it impact performance in the IID setting when lambda is set to 1 instead? I think this is likely a substantial restriction of the proposed work in case we do not know the data heterogeneity of the clients. Could the authors comment more on this? Any practical guidelines on choosing lambda here? Fig. 1 suggest a range of (0,1) for lambda but does it apply generically or only to CIFAR data?

Another look at the supplement results in Table 6 shows that the performance is also sensitive somewhat to the choice of the architecture. How would this impact the default range of good values for lambda?

EXPERIMENT

The experiment is quite extensive & interesting. But I do have a few follow-up questions:

In Tables 3 and 4, what are FedEMA's performance on IID setting when lambda = 1.

How many runs were used to produce the average accuracy and standard deviation?

What is the exact non-IID setting used in Tables 3 and 4?

There seems to be a decrease in terms of accuracy in CIFAR-100 when one moves from non-IID to IID settings. This is the opposite of what we saw in CIFAR-10 settings. Could the authors comment on this?

It seems some baselines such as FedMoCoV1 and FedMoCoV2 were missing from Table 7. Could the authors provide the missing info?

**Summary Of The Paper:**

This paper investigates a generic federated SSL recipe that applies FedAvg to a range of existing SSL works including SimCLR, MoCo, BYOL and SimSiam. Each of these SSL blocks comprises two encoding networks, including an online net and a target net. The two nets were trained via optimizing a similarity loss such that the distance between encodings of different augmented versions (e.g. rotation) of the same input is small and vice versa. Depending on the specific SSL block, the parameterization of the two nets might be identical (SimCLR, SimSiam) or not (MoCo, BYOL). At each iteration, the online net of each client is uploaded to a server to be averaged. Then, the (global) averaged version is sent back to each client. Each client then reset its online net as a weighted average of the global and local version of the online net (to account for potential data heterogeneity), and continue updating the SSL block via gradient updates and so on.

Here, the main contribution of the paper is a series of ablation studies (Table 1, Figure 2, Table 2, Figure 3) that measure the isolated impact of several fundamental components of the aforementioned SSL blocks (e.g., SimCLR, SimSiam, MoCo and BYOL). This includes the predictor, stop-gradient, exponential moving average (EMA), non-identical online and target encoding net. The results indicate that BYOLD contains all the essential elements (predictor, EMA and non-identical online & target net & no stop-gradient) so the federated SSL recipe is rotated towards having BYOL as the model choice. Then, a customized, divergence-aware EMA is proposed for this recipe that is shown to improve substantially over existing FL-SSL baselines (Tables 3-4; Tables 6-7). There is also a bit of ablation study that measures the isolated impact of the new EMA which is also substantial as shown in Figure 5.

**Summary Of The Review:**

This is an empirical paper. The key contribution is an investigative, ablation studies that build up intuitions and practical guidelines to fine-tune the generic federated SSL framework. The fine-tuned version is shown to have significant performance improvement. But on another note, this success might be dependent on a correct choice of lambda which is only possible if one knows the data heterogeneity of the clients. Perhaps this is a key restriction of the method. It would be great if the authors can expand on this during the rebuttal.

---

> ### Author Response · Authors · 2021-11-20
> **Reply to Reviewer wF5G (1/2)**
>
>
> We thank the reviewer for the valuable comments.
>
> **Q1**: How to choose $\lambda$? Range values of $\lambda$.
>
> **A1.1**: Thank you for raising this important question. Inspired by your question, we further propose a practical \textit{autoscaler} to calculate a personalized $\lambda_k$ for each client $k$ automatically. The formula is $\lambda_k = \frac{\tau}{||W_g^{r+1} - W_k^r||}$, where $\tau \in [0, 1]$ is the expected value of $\mu$ at round $r$. We calculate $\lambda_k$ only once at the earliest round $r$ that client $k$ is sampled for training. When the same set of clients are sampled for training, $\lambda_k = \frac{\tau}{||W_g^1 - W_k^0||}$ is calculated at round $r=1$. Since model divergence is larger at the start of training (Figure 6), it is practical to choose larger $\tau \in [0.5, 1)$; $\tau = 1$ is not considered because only local knowledge is used when $\tau = 1$. We have provided ablation studies on $\tau$ in Figure 10(a) in Appendix C. Autoscaler achieves comparable, sometimes better results, than choosing $\lambda$ manually, as compared in the folloing three tables. We use $\tau = 0.7$ by default in experiments.
>
> Table 1: Top-1 accuracy comparison on linear evaluation protocol.
>
> | Methods | CIFAR-10 IID | CIFAR-10 Non-IID | CIFAR-100 IID | CIFAR-100 Non-IID |
> | --- | --- | --- | --- | --- |
> | FedEMA ($\lambda=0.8$) | 85.59 $\pm$ 0.25 | 82.77 $\pm$ 0.08 | 57.86 $\pm$ 0.15 | 61.21 $\pm$ 0.54 |
> | FedEMA (autoscaler) | 86.26 $\pm$ 0.26 | 83.34 $\pm$ 0.39 | 58.55 $\pm$ 0.34 | 61.78 $\pm$ 0.14 |
>
> Table 2: Top-1 accuracy comparison on semi-supervised learning evaluation.
>
> | Methods | CIFAR-10 IID | CIFAR-10 Non-IID | CIFAR-100 IID | CIFAR-100 Non-IID |
> | --- | --- | --- | --- | --- |
> | FedEMA ($\lambda=1$) | 72.78 $\pm$ 0.66 | 79.01 $\pm$ 0.30 | 32.49 $\pm$ 0.22 | 49.82 $\pm$ 0.36 |
> | FedEMA (autoscaler) | 73.44 $\pm$ 0.22 | 79.49 $\pm$ 0.34 | 33.04 $\pm$ 0.23 | 50.48 $\pm$ 0.11 |
>
> Table 3: Top-1 accurarcy comparison on various batch sizes on the non-IID setting of CIFAR-10 dataset. We use λ = 0.4 for B = 32 and λ = 0.6 for B = 64.
>
> | Methods | B = 32 | B = 64 | B = 128 |
> | --- | --- | --- | --- |
> | FedEMA (manual $\lambda$) | 79.55 | 81.31 | 82.77 | 82.70 | 79.93 |
> | FedEMA (autoscaler) | 79.06 | 82.18 | 83.34 |
>
> **A1.2**: The range values of $\lambda$ depend on the characteristics of data and the hyper-parameters (e.g., local epoch) of FL settings. A practical way to tune $\lambda$ manually is to understand the divergence by running the algorithm for several rounds and choose the $\lambda$ that scales $\mu$ to (0.5, 1). Nevertheless, we recommend using autoscaler. We have also revised the discussion on Scaler in Section 5.2 in the manuscript to reflect these comments to prevent misunderstanding.
>
> **Q2**: Impact of network architecture on $\lambda$.
>
> **A2**: The architecture does not significantly impact the range of good values of $\lambda$, compared with characteristics of data and the hyper-parameters (e.g., local epoch) of FL settings. We use the same value of $\lambda$ for both ResNet-18 and ResNet-50 in the paper. Autoscaler mentioned in Q1 could also address this concern by calculating the $\lambda$ automatically.

---

> > ### Author Response · Authors · 2021-11-20
> > **Reply to Reviewer wF5G (2/2)**
> >
> > EXPERIMENT
> >
> > **Q3**: In Tables 3 and 4, what are FedEMA's performances on the IID setting when λ = 1?
> >
> > **A3**: For linear evaluation on IID setting in Table 3, the performances of FedEMA are 84.06% on CIFAR-10 and 57.54% on CIFAR-100 when λ = 1. They are close to the results in Table 3 (85.59% for CIFAR-10 and 57.86% for CIFAR-100) when λ = 0.8. Results in Table 4 are on non-IID settings for semi-supervised learning protocol, where λ = 1. We provide the results of IID settings for λ = 1 in Table 4 below.
> >
> > Table 4: Top-1 accuracy comparison on IID settings under semi-supervised learning evaluation.
> >
> > | Methods | CIFAR-10 1% | CIFAR-10 10% | CIFAR-100 1% | CIFAR-100 10% |
> > | --- | --- | --- | --- | --- |
> > | FedBYOL | 80.23 | 82.00 | 26.66 | 44.61 |
> > | FedEMA | 80.59 | 82.47 | 31.78 | 47.79 |
> >
> > **Q4**: How many runs were used to produce the average accuracy and standard deviation?
> >
> > **A4**: Due to the large number of experiments in the paper, we conduct 3 runs per experiment to calculate the average and standard deviation.
> >
> > **Q5**: What is the exact non-IID setting used in Tables 3 and 4?
> >
> > **A5**: As mentioned in the implementation details in Section 3.3, the default setting of non-IID is 2 (20) classes per client for CIFAR10 (CIFAR100). Table 3 and 4 use this default setting.
> >
> > **Q6**: Why accuracy decreases from non-IID to IID on CIFAR-100, but not on CIFAR-10?
> >
> > **A6**: Thank you for raising the question. Results on CIFAR-10 are intuitive as the performance is normally better on the IID setting than the non-IID setting. In contrast, the results on CIFAR-100 show that the non-IID is better than the IID setting. Table 5 below compares the number of classes and images per class among these settings. Our interpretation is that the non-IID setting of CIFAR-100 has enough (20) classes per client for learning generic representations, and the IID setting of CIFAR-100 has only 100 images per class, which may not be enough for learning good representations.
> >
> > Table 5: Details of CIFAR-10 and CIFAR-100 datasets under IID and non-IID settings.
> >
> > | Datasets | Settings | Classes per client | Images per class |
> > | --- | --- | --- | --- |
> > | CIFAR-10 | IID | 10 | 1000 |
> > | CIFAR-10 | Non-IID | 2 | 5000 |
> > | CIFAR-100 | IID | 100 | 100 |
> > | CIFAR-100 | Non-IID | 20 | 500 |
> >
> > **Q7**: It seems some baselines such as FedMoCoV1 and FedMoCoV2 were missing from Table 7.
> >
> > **A7**: We have run these experiments (FedSimCLR, FedSimSiam, FedMoCoV1, and FedMoCoV2) and added them in Table 7 in Appendix. C.

---

### Official Review · Reviewer_48jX · 2021-11-06

**Correctness:** 3
**Technical Novelty And Significance:** 2
**Empirical Novelty And Significance:** 2
**Recommendation:** 3
**Confidence:** 5

**Main Review:**

* Strengths
1. Unsupervised FL is a well-motivated and timely topic.


* Weaknesses
1. This paper is a simple combination of BYOL and FedAvg. I cannot see significant novelty in both BYOL and FedAvg-like algorithm. Especially, its relationship to ICCV 2021 paper FedBYOL [1] is not fully discussed.

2. Why is SimSaim [2] not used as the SSL framework? It has better interpretability in optimization; it requires smaller batch size; it doesn't require moving averaging. I believe the performance of SimSiam would be better than BYOL in FL context.

3. The definition of unlabeled non-I.I.D. dataset is not clear. According to Section 3.3, the authors use CIFAR-10 and CIFAR 100 to partition the dataset in an unbalanced manner with respect to the number of classes in each client. This is contradicted to the unsupervised assumption because we don't know users' labels, then it seems we cannot claim their data is non-I.I.D. in terms of the label. A careful definition of "unsupervised non-I.I.D. data" is required in this work.

4. Note that FL clients should not be stateful. When numerous clients exist, we need to do client sampling for a new round of training. The newly sampled client do not have any cached states to conduct adaptive optimizer or moving averaging-like stateful algorithms [4]. Please explain how Algorithm 1 can work in cross-device FL.

5. The batch size is a challenge in FL context since FL clients normally are resource-constrained. Batch size = 128 is too big for smartphones.

6. The source code is not provided for reproducibility. I doubt some experiment results and would like to reproduce some results to confirm the efficacy.

7. It would be better to discuss the privacy/security concern of the proposed FedSSL framework.

---
[1] Collaborative unsupervised visual representation learning from decentralized data. ICCV 2021.

[2] Exploring Simple Siamese Representation Learning. CVPR 2021.

[3] Federated Reconstruction: Partially Local Federated Learning. NeurIPS 2021.


**Summary Of The Paper:**

This paper combines self-supervised learning framework BOYL and FedAvg to improve the performance of unlabeled non-IID datasets.

**Summary Of The Review:**

I suggest NOT accepting this paper at the current version, given that a few issues mentioned above are critical.

---

> ### Author Response · Authors · 2021-11-20
> **Reply to Reviewer 48jX (1/2)**
>
> We thank the reviewer for the feedback and address the detailed comments below:
>
> **Q1**: Novelty and relationship to [1].
>
> **A1**: Thank you for raising the concern. We would like to clarify that our methods are not a simple combination of BYOL and FedAvg. Although ICCV 2021 paper [1] proposes a new method (FedU) based on BYOL, they do not shed light on why BYOL works best. Since SSL methods are evolving rapidly and new methods (like SimSiam [2], MoCoV2, SimCLR) are emerging, it is crucial to understand the fundamental components of FedSSL. In this paper, we first introduce a generalized FedSSL framework that embraces existing SSL methods and has flexibility catering to future methods. Then, we took an investigative approach to deeply investigate the fundamental components of FedSSL to build up intuitions and practical guidelines for the generic FedSSL framework. Based on these insights, we propose a new model update approach, FedEMA, which improves the performance substantially. We have revised the Related Work section with these discussions. Besides, we would like to point out that FedEMA outperforms [1], as shown in Table 3, 4, 5, and 7 in the manuscript.
>
> **Q2**: Why is SimSaim [2] not used as the SSL framework?.
>
> **A2**: We agree that the architecture of SimSiam is simpler and have investigated SimSiam in the FedSSL framework. However, FedSimSiam substantially underperforms FedBYOL as shown in Table 1 & 2 in the paper. FedBYOL also outperforms FedSimSiam when using a smaller batch size (B=32), as shown in the table below. Experiments are run with ResNet-18 on the non-IID CIFAR-10 dataset.
>
> Table 1: Top-1 accuracy comparison on smaller batch size (B = 32).
>
> | Model | B = 32 |
> | --- | --- |
> | FedSimSiam | 71.44 |
> | FedBYOL | 72.90 |
> | FedEMA (ours) | 79.55 |
>
>
> **Q3**: Definition of "unsupervised non-I.I.D. data".
>
> **A3**: Thank you for raising the concern. We do not define the non-IID setting of CIFAR-10/100 datasets by ourselves but follow the standard way as [1][3][5][6]. Even in the unlabeled scenario, the underlying non-IID characteristic of the data still exists.
>
> **Q4**: Note that FL clients should not be stateful.
>
> **A4**: Thank you for raising the concern. Actually, this paper focuses on cross-silo FL rather than cross-device FL and we have revised the manuscript to clarify this point. We have also refined Algorithm 1 to handle client sampling on every training round. Specifically, when clients are newly sampled, we update their online and target encoders with the global model; when clients are sampled in the previous round, we update only the online encoders with EMA. In definition, cross-device FL assumes there are millions of stateless clients that might participate in training just once. While due to the constraints of experimental settings, the majority of studies conduct experiments with at most hundreds of clients. FedEMA can work under such experimental settings by caching states of clients in the server. When the number of clients scales to millions, FedEMA degrades to FedBYOL that updates both encoders --- without keeping any local states. We have included the discussion on cross-silo FL and cross-device FL in Appendix B.3.

---

> > ### Author Response · Authors · 2021-11-20
> > **Reply to Reviewer 48jX (2/2)**
> >
> >
> > **Q5**: Batch size is too big.
> >
> > **A5**: Thank you for raising the concern. Our method works across different batch sizes. Table 2 below shows that smaller batch sizes (B = 16 or B = 32) achieve comparable results as B = 128 on the non-IID CIFAR-10 dataset.  Besides, some [5][7] FL studies also consider batch sizes larger than 100 as FL can also be used in more powerful edge devices other than smartphones.
> >
> > Table 2: Top-1 accuracy comparison on smaller batch sizes (B = 16 and 32), with learning rate adjusted proportionally to the batch size.
> >
> > | Methods | B = 16, lr = 0.004 | B = 32, lr = 0.008 | B = 128, lr = 0.032 |
> > | --- | --- | --- | --- |
> > | FedSimSiam | 77.48 | 78.52 | 76.27 |
> > | FedBYOL | 78.96 | 79.54 | 79.44 |
> > | FedEMA | 80.22 | 80.05 | 82.77 |
> >
> > **Q6**: Code for reproducibility.
> >
> > **A6**: We are under the internal process to open-source the code, while we have provided the core implementation of the source code for review in another comment only to reviewers.
> >
> > **Q7**: Discussion on privacy/security concern.
> >
> > **A7**: Thank you for raising this concern. Our proposed FedSSL framework and FedEMA method do not introduce extra privacy/security concerns compared with the standard FL scenario (FedAvg), thus, we do not provide a section to specifically discuss them.
> >
> > References:
> >
> > [1] Collaborative unsupervised visual representation learning from decentralized data. ICCV 2021.
> >
> > [2] Exploring Simple Siamese Representation Learning. CVPR 2021.
> >
> > [3] HeteroFL: Computation and communication efficient federated learning for heterogeneous clients. ICLR 2021.
> >
> > [4] Federated Reconstruction: Partially Local Federated Learning. NeurIPS 2021.
> >
> > [5] Federated Semi-Supervised Learning with Inter-Client Consistency & Disjoint Learning. ICLR 2021.
> >
> > [6] Federated Learning with Non-IID Data. ArXiv 2018.
> >
> > [7] Performance optimization of federated person re-identification via benchmark analysis. ACMMM 2020.
> >
> > [8] Kairouz, Peter, et al. "Advances and open problems in federated learning." arXiv preprint arXiv:1912.04977 (2019).

---

> > ### Comment · Reviewer_48jX · 2021-11-22
> > **still not ready for acceptance at the current stage**
> >
> > Dear Authors,
> >
> > Here is my further response.
> >
> > Q1: I still cannot find significant differences and novelty compared to the ICCV 2021 paper.
> >
> > Q2: Your results are not convincing. At Arxiv, I can find another concurrent work that can obtain much higher accuracy in CIFAR-10 + ResNet-18 using FedSimSiam (https://arxiv.org/pdf/2110.02470.pdf, Table 1, 84.7%). Your is only 71.44%. I don't think the gap should be so large. This demonstrates that FedSimSiam works much better than FedBYOL and FedEMA. I don't believe FedSimSiam works worse than FedBYOL. Please have a solid hyper-parameter searching.
> >
> > Q4: The proposed method is stateful, thus it cannot work on cross-device FL. Then you admit that it should be downgraded to cross-silo FL. However, in cross-silo FL, the label won't be a serious issue given that each organization (e.g., hospital, bank, etc.) can label their data for sure. Or at least, it should be semi-supervised FL. As such, I think the motivation is not strong at cross-silo FL.
> >
> > Given these reasons, I think the FL community should NOT accept such a method as a baseline for future research.

---

> > > ### Author Response · Authors · 2021-11-24
> > > **Thank you for your further comments**
> > >
> > > Thank you for your further feedback. These questions seem to be the same as [these ones](https://openreview.net/forum?id=oVE1z8NlNe&noteId=YuWbAQQOhdX) and we have provided detailed replies there.

---

### Author Response · Authors · 2021-11-21
**Details of changes made in the updated version**

We thank all reviewers for the time and expertise they have invested in these reviews. We have addressed all comments and improved our paper according to your suggestions. Here, we would like to point out the changes made during this revision.

1. We positioned the paper to primarily focus on cross-silo FL (in the introduction) and added discussion on cross-silo Fl and cross-device FL in Appendix B.3.

2. We proposed a new *autoscaler* method to calculate a personalized $\lambda_k$ for each client $k$ automatically. We added experiments in Table 3 and 4, provided ablation study in Figure 10(a), updated Table 10, to demonstrate the effectiveness of this method.

3. We refined Algorithm 1 to include the proposed *autoscaler* and support subsampling of clients.

4. We added experiments of random client sampling in Table 6 in Appendix C.

5. We added more baseline experiments of FedSSL methods (FedSimSiam, FedMoCoV1, FedMoCoV2, FedSimCLR) for semi-supervised learning evaluation in Table 7 in Appendix C.

6. We further clarify the relationship of our paper with the ICCV 2021 paper in Section 2.

7. We provided the update frequency of the momentum encoder of local training and the definition of $W$ in Section 3.

8. We shifted the discussion on communication and computation cost to Appendix C due to space constraints.

---

> ### Comment · Reviewer_48jX · 2021-11-22
> **significant concerns**
>
> Dear Authors,
>
> Here is my further response.
>
> 1. I still cannot find significant differences and novelty compared to the ICCV 2021 paper.
>
> 2. Your results are not convincing. At Arxiv, I can find another concurrent work that can obtain much higher accuracy in CIFAR-10 + ResNet-18 using FedSimSiam (https://arxiv.org/pdf/2110.02470.pdf, Table 1, 84.7%). Your is only 71.44%. I don't think the gap should be so large. This demonstrates that FedSimSiam works much better than FedBYOL and FedEMA. I don't believe FedSimSiam works worse than FedBYOL. Please have a solid hyper-parameter searching.
>
> 3. The proposed method is stateful, thus it cannot work on cross-device FL. Then you admit that it should be downgraded to cross-silo FL. However, in cross-silo FL, the label won't be a serious issue given that each organization (e.g., hospital, bank, etc.) can label their data for sure. Or at least, it should be semi-supervised FL. As such, I think the motivation is not strong at cross-silo FL.
>
> Given these reasons, I think the FL community should NOT accept such a method as a baseline for future research.

---

> > ### Comment · Reviewer_Nu54 · 2021-11-23
> > **A comment about labels in the cross-silo setting**
> >
> > I disagree about labels not being an issue in the cross-silo setting. Annotating e.g. medical image data can be a significant time and financial investment, and requires highly trained specialists. Unsupervised methods in the cross-silo setting absolutely have their justification.

---

> > > ### Comment · Reviewer_48jX · 2021-11-23
> > > **that's why I said semi-supervised FL is more practical in cross-silo FL**
> > >
> > > Please note that the proposed framework cannot support semi-supervised FL.

---

> > > > ### Comment · Reviewer_Nu54 · 2021-11-23
> > > > **I am confused**
> > > >
> > > > The authors position their method (and evaluation) mostly as unsupervised work, although they do include extensive results on a semi-supervised evaluation setting (Table 4 and Table 7). I agree that the evaluation is focussed on the case of labelled dataset being present in a central location (thus not FL). I imagine that the MLP could be trained in the usual supervised FL way based on the encoder representations, once trained. What 'support for semi-supervised FL' do you see as missing?

---

> > > > > ### Comment · Reviewer_48jX · 2021-11-30
> > > > > **further response**
> > > > >
> > > > > I get your point. But the stateful design really downgrades this paper a lot.
> > > > >
> > > > > After reading all replies, I still have too many concerns: 1) the novelty, 2) the stateful optimizer, 3) the unnecessary EMA and the lack of explainability for it; 4) the results of the experiment are unconvincing and NOT reproducible.
> > > > >
> > > > > As such, I still recommend rejection.

---

> > ### Author Response · Authors · 2021-11-24
> > **Reply to Reviewer 48jX (1/2)**
> >
> > **Q1**: I still cannot find significant differences and novelty compared to the ICCV 2021 paper.
> >
> > **A1**: We would like to further clarify the differences and novelty by comparing with ICCV 2021 paper in the Table below. Apart from the newly proposed model update method (FedEMA) in Section 4, we would like to invite the reviewer to revisit Section 3 where we provide important insights into FedSSL, which is not studied in the ICCV 2021 paper but builds up the intuition and practical guidelines for our new method.
> >
> > | | ICCV 2021 Paper| Ours |
> > | --- | --- | --- |
> > | Evaluated SSL method | BYOL | SimCLR, MoCoV1, MoCoV2, SimSiam, BYOL |
> > | Insights on | Communication protocol and predictor update| Fundamental components of SSL methods, including predictor, stop-gradient, momentum update, online and target encoder |
> > | Model update method | Update the predictor using either local or global predictor, determined by whether model divergence is larger or smaller than a threshold.| Update both encoder and predictor using EMA of global and local models, where the decay rate of EMA can be calculated automatically. |
> >
> > **Q2**: Your results are not convincing. Concurrent work at Arxiv [1] obtains 84.7%, but yours is 71.44%.
> >
> > **A2.1**: Thank you for raising the concern. We would like to justify that the results are not directly comparable as they are under different experiment settings using different hyper-parameters.
> > Otherwise, we can even achieve 86.09% of FedBYOL in Figure 9 in Appendix C.
> >
> > The mentioned paper [1] achieves 84.7% by running $R = 800$ rounds with local epoch $E = 1$, batch size $B = 256$, and learning rate $\eta = 0.1$ on non-IID CIFAR-10 dataset sampled by Dirichlet distribution with $\alpha = 0.1$ (Appendix E.2). In contrast, our result of 71.44% is obtained with $B = 32$ to show that FedSimSiam *does not* work better with too small batch sizes. To facilitate analysis, we provide Table 1 below, which is run with $E = 1$, $B = 128/256$, and $R = 800$ on two different non-IID settings -- 2 classes per client (ours) and Dirichlet sampling with $\alpha = 0.1$ (*Dir($\alpha=0.1$)*). We are able to obtain results of FedSimSiam fairly similar to [1] (only 0.5% difference). We tried our best to match the experiment settings, but we could not obtain enough information such as the total number of clients they used. There are some other variances such as learning rate scheduler and model size (details in Appendix B.2). Nevertheless, we would like to point out that FedBYOL consistently outperforms FedSimSiam in different rounds under different settings.
> >
> > Table 1: Top-1 accuracy comparison of FedSimSiam and FedBYOL using similar hyper-parameters as [1].
> >
> > | Methods | Non-IID | Batch Size | r=100 | r=200 | r=300 | r=400 | r=500 | r=600 | r=700 | r=800 |
> > | --- | --- | --- | --- | --- | --- | --- | --- | --- | --- | --- |
> > | FedSimSiam | 2 classes per client | B=256 | 59.02 | 74.20 | 77.31 | 79.18 | 80.46 | 81.57 | 82.14 | 82.00 |
> > | FedBYOL | 2 classes per client | B=256 | 65.91 | 76.03 | 79.19 | 81.07 | 82.06 | 82.32 | 82.34 | 82.44 |
> > | FedSimSiam | 2 classes per client | B=128 | 69.68 | 75.09 | 76.76 | 78.44 | 80.40 | 82.00 | 82.02 | 81.80 |
> > | FedBYOL | 2 classes per client | B=128 | 64.51 | 75.31 | 79.62 | 81.21 | 82.72 | 83.65 | 84.45 | 84.38 |
> > | FedSimSiam | Dir(0.1) | B=128 | 72.34 | 76.09 | 78.52 | 80.47 | 82.14 | 83.33 | 83.57 | 83.54 |
> > | FedBYOL | Dir(0.1) | B=128 | 72.80 | 77.55 | 80.00 | 82.20 | 83.56 | 84.54 | 85.27 | 85.39 |
> >
> > **A2.2**: Moreover, we followed your suggsetion to further conduct hyper-parameter searching for FedSimSiam. We evaluate batch size $B = {128, 256}$ and learning rate $\eta = {0.01, 0.032, 0.1, 0.3}$ (similar to [1]) for 100 rounds in Table 2 below; FedBYOL still outperforms FedSimSiam. Given these results and the results in the paper, we are confident that FedBYOL is better than FedSimSiam in our setting.
> >
> > Table 2: Hyperparameter search for FedSimSiam.
> >
> > | Methods | Batch Size | lr = 0.01 | lr = 0.032 | lr = 0.1 | lr = 0.3 |
> > | --- | --- | --- | --- | --- | --- |
> > | FedSimSiam | B = 256 | 71.36 | 75.37 | 21.56 | 10.00 |
> > | FedSimSiam | B = 128 | 78.36 | 76.27 | 10.00 | 10.00 |
> > | FedBYOL | B = 128 | 80.96 | 79.44 | - | - |
> >
> > **A2.3**: Lastly, we would like to point out that the nice property of SimSiam, *requiring smaller batch size*, is claimed in the context of centralized training in their original paper [2] and Appendix A of [1]; SimSiam achieves comparable/better results with $B = 256$, while other SSL methods use $B = 4096$. It does not suggest that FedSimSiam works better with even smaller batch sizes; on the contrary, smaller batch sizes (e.g., $B = 32$) would degrade performance, as shown in [reply to the previous Q2](https://openreview.net/forum?id=oVE1z8NlNe&noteId=aFQq9U-Uqm0)  and Figure 4 of [1]. Also, it does not suggest that other SSL methods would not work well in the context of FL using batch size like $B = 128/256$, as shown in Table 1 and Table 6 in the manuscript.

---

> > > ### Author Response · Authors · 2021-11-24
> > > **Reply to Reviewer 48jX (2/2)**
> > >
> > >
> > > **Q3**: The proposed method is stateful, thus it cannot work on cross-device FL. Then you admit that it should be downgraded to cross-silo FL. However, in cross-silo FL, the label won't be a serious issue given that each organization (e.g., hospital, bank, etc.) can label their data for sure. Or at least, it should be semi-supervised FL. As such, I think the motivation is not strong at cross-silo FL.
> > >
> > > **A3**: Thank you for raising the concern. Firstly, we would like to point out that the paper is positioned to primarily focus on cross-silo FL as suggested by reviewer Nu54, while it can also work in cross-device FL as discussed in [reply to the previous Q4](https://openreview.net/forum?id=oVE1z8NlNe&noteId=aFQq9U-Uqm0) (and Appendix B.3). Secondly, annotating data is time-consuming, laborious, and expensive, which is also a crucial problem in cross-silo FL. Some examples are provided by Reviewer Nu54 in the [comment](https://openreview.net/forum?id=oVE1z8NlNe&noteId=o0eaIRb9kZy). Thirdly, semi-supervised FL is an interesting topic, while we focus on the purely unsupervised methods; Organizations could have the resources to label data, but we are seeing large companies like Facebook and Google are also interested in purely unsupervised methods. For example, MoCoV1/V2 [3] and SimCLR [4] are done by Facebook and Google, respectively.
> > >
> > > References:
> > >
> > > [1] SSFL: Tackling Label Deﬁciency in Federated Learning via Personalized Self-Supervision. Arxiv 2021.
> > >
> > > [2] Exploring Simple Siamese Representation Learning. CVPR 2021.
> > >
> > > [3] Momentum Contrast for Unsupervised Visual Representation Learning. CVPR 2020.
> > >
> > > [4] A Simple Framework for Contrastive Learning of Visual Representations. ICML 2020.

---

> > > ### Comment · Reviewer_48jX · 2021-11-30
> > > **still no significant contribution and novelty**
> > >
> > > For the points you highlighted about ICCV paper, it doesn't make sense to me:
> > > 1. "Fundamental components of SSL methods, including predictor, stop-gradient, momentum update, online and target encoder" --- There are not from your contribution. "predictor, stop-gradient, momentum update, online and target encoder" are all from BOYL paper.
> > > 2. doing EMA in FL setting doesn't make sense to me, not elegant like the FedSimSiam baseline. I don't believe the results given in the source code are not reproducible.

---

> > > > ### Author Response · Authors · 2021-12-01
> > > > **Reply to Reviewer 48jX**
> > > >
> > > > **Q1**: "Fundamental components of SSL methods, including predictor, stop-gradient, momentum update, online and target encoder" --- There are not from your contribution. "predictor, stop-gradient, momentum update, online and target encoder" are all from BOYL paper.
> > > >
> > > > **A1**: These components are the building blocks of trending SSL methods, as shown in Figure 8 in the manuscript. Moreover, we have to emphasize that our contribution is not proposing these components but to shed light on deep **insights** of their impact in the **FedSSL** framework, which is not studied previously.
> > > >
> > > > **Q2**: doing EMA in FL setting doesn't make sense to me, not elegant like the FedSimSiam baseline. I don't believe the results given in the source code are not reproducible.
> > > >
> > > > **A2**: *Elegant* is a very subjective description for scientific discussion. Once again, we would like to clarify that we will open-source the code after the internal process and have provided the core implementation for review. We also would like to point out that it is absurd to discredit a paper just because the source code is not provided.

---

### Decision · Program_Chairs · 2022-01-20

**Decision:**

Accept (Poster)

**Comment:**

The paper focuses on self-supervised learning (SSL) in the federated learning setting (FedSSL). Research in this area is timely and of significance. The authors phrase their work as primarily being an empirical study providing insights into the building blocks of FedSSL. The evaluation in the paper is quite thorough and the authors have been active in a detailed exchange regarding questions raised in the reviews. I would encourage the authors to fully implement the changes they promised into the revised manuscript and work towards timely release of open-source code. (I appreciate internal policies of various institutions, but I do agree with the reviewers that it is more important that the code and experimental details be made public for papers such as this one, compared to some other papers.) I have chosen to disagree with some of the concerns raised in one of the reviews, in particular, I do agree with the authors that insights into the building blocks through empirical studies is a significant contribution, and also that FedEMA is a novel contribution. The discussion on this forum will remain for interested readers to come to their own conclusions about the relative performance of various methods.